# Compress–Add–Smooth: Fixed-Budget Temporal Compression of Density-Valued Streams

## Abstract

We study fixed-budget temporal memory for streams of probability distributions. The proposed representation is a piecewise-linear density protocol on a replay interval $[0, 1]$: recent experience is stored near $t = 1$, older experience is represented by intermediate-time marginals, and new experience is incorporated by a deterministic *Compress–Add–Smooth* (CAS) recursion. In the Gaussian-mixture instantiation considered here, each protocol node stores a labeled $K$-component Gaussian mixture in $d$ dimensions, and each daily update costs $O(LKd^2)$ arithmetic operations for a fixed temporal budget of $L$ segments. No gradient training, stored raw examples, or neural-network replay model is required for the CAS update itself.

The contribution is best viewed as an analytically transparent model of temporal density compression and forgetting, rather than as a complete replacement for end-to-end neural continual-learning systems. Forgetting arises from one explicit lossy operation: rebinning an $(L + 1)$-segment protocol back to $L$ segments. Under moment-based diagnostics on smooth synthetic Gaussian-mixture streams and an MNIST PCA latent-space illustration, the forgetting curves exhibit a recent-memory plateau followed by a sigmoid-like degradation, and the retention half-life is approximately linear in $L$, $a_{1/2} \approx cL$. The coefficient $c$ should be interpreted as source-, representation-, and metric-dependent; in the present experiments it varies with drift speed and with the chosen diagnostic. We also distinguish the deterministic CAS density protocol from an optional stochastic realization: given a prescribed density path, an SDE drift can be reconstructed from the Fokker–Planck equation, but SDE trajectory replay is not used by the CAS memory update and requires separate validation.

## 1 Introduction

**Problem setting.** Many sequential systems — controllers, robots, sensor nodes, simulators, and learning agents — encounter a stream of changing operating conditions. In this paper we abstract each day's experience as a probability distribution $q^{(m)}$ over a physical or latent state space. The memory problem is then to maintain a fixed-size representation of the distribution-valued history

$$q^{(1)}, q^{(2)}, \ldots, q^{(n)},$$

so that past distributions can be queried or replayed approximately without storing the full stream.

This abstraction is motivated by continual learning, replay, and resource-limited inference, but it is deliberately narrower than full neural continual learning. Classical continual-learning failures arise when sequential gradient updates overwrite parameters needed for older tasks McCloskey and Cohen (1989); French (1999). Here there is no shared classifier or policy being trained. Instead, we isolate a complementary mechanism: *forgetting by temporal compression* of a density-valued stream. The goal is to make this mechanism explicit enough to measure, analyze, and compare.

**CAS density protocol.** The memory is a piecewise-linear density protocol $p_t$, $t \in [0,1]$. The terminal marginal $p_1$ represents the most recent day. Earlier days are queried at geometrically decaying readout times. Incorporating a new distribution is a deterministic three-step recursion: *Compress* the old protocol from $[0,1]$ to $[0, L/(L+1)]$; *Add* the new distribution at $t = 1$; and *Smooth* by rebinning the resulting $(L+1)$-segment protocol back to the fixed $L$-segment grid. Only the smoothing/rebinning step is lossy.

We instantiate this construction with labeled Gaussian mixtures. Each node stores $K$ mixture weights, means, and covariance matrices in dimension $d$, so the fixed protocol stores $O(LKd^2)$ floating-point numbers and one update costs $O(LKd^2)$ arithmetic operations. The method can be applied to an unlabeled fitted mixture after an explicit component-alignment step; without such alignment, componentwise interpolation is not permutation invariant. We make this labeled-mixture assumption explicit below.

**Deterministic memory versus optional stochastic realization.** The CAS update is a deterministic operation on density parameters. It does not require simulating trajectories, solving an SDE, or reconstructing a drift. Separately, once a density path $p_t$ has been specified, one can ask whether there exists an Itô process whose one-time marginals are exactly $p_t$. Appendix A gives such a Fokker–Planck reconstruction. This optional SDE realization is useful for studying temporally coherent replay trajectories, but it is not part of the fixed-memory update and thus does not contribute to the density-level CAS algorithm.

**Relation to continual learning and replay.** The method is closest in spirit to streaming summaries, multiresolution histories, and compression-based replay. It differs from diffusion-based generative replay methods Gao and Liu (2023); Jodelet et al. (2023); Meng et al. (2024); Kim et al. (2024); Liang et al. (2024); He et al. (2024); Hu et al. (2025), where a learned generator produces samples for a downstream model. In the present paper the compressed density protocol itself is the object of study. Downstream control or classification performance is therefore a separate empirical question, not something established by the forgetting curves alone.

**Summary of current evidence.** We evaluate CAS on synthetic Gaussian and Gaussian-mixture streams and on an MNIST PCA latent-space illustration. The current experiments support the following conclusions:

1. The moment-based forgetting curves have a two-regime form: a low-error plateau for recent memories followed by a sigmoid-like degradation with age.
2. For the tested smooth streams and diagnostics, the retention half-life grows approximately linearly with the segment budget, $a_{1/2} \approx cL$. The coefficient $c$ is not universal: it depends on the source dynamics, the representation, and the evaluation metric.
3. Increasing the number of mixture components $K$ did not change the moment-based half-life in the controlled labeled-mixture experiments. This is interpreted as a property of the tested setting and metric.
4. The dominant error channel depends on the stream: mean errors dominate in the synthetic mean-drift examples, while covariance and weight-related effects are more visible in the MNIST latent-space construction.
5. The MNIST protocol can be decoded as a smooth marginal "movie" across protocol time. This is a visualization of the stored density protocol; validating actual SDE trajectory replay is a separate experiment.
6. A lightweight MNIST latent replay task converts replayed Gaussian mixtures into class-posterior predictions. This task exposes a limitation of the uniform CAS grid: although CAS preserves a coherent density protocol, task-level posterior accuracy and NLL are worse than for several memory-matched streaming buffers that store or adaptively place daily GM states.

**Paper outline.** Section 2 defines the CAS recursion. Section 3 identifies the single lossy operation responsible for forgetting. Section 4 defines the current moment-based and density-sensitive metrics. Sections 5–7 report controlled Gaussian-mixture experiments and memory-matched streaming baselines. Section 8 gives the MNIST latent-space illustration and the latent replay classification proxy. Section 9 discusses the empirical linear retention law and the optional SDE realization. Section 10 concludes. Appendix A gives the density-to-drift construction.

## 2    The Compress–Add–Smooth Framework

The agent maintains a deterministic density protocol on the fixed replay interval $[0, 1]$: the marginal at $t = 1$ represents the current day, and earlier portions of the interval encode progressively older information. Incorporating a new day is a three-step recursion — compress, add, smooth — carried out entirely within a chosen parameterised density class. We illustrate it on labeled Gaussian mixtures (GM), where all operations reduce to linear algebra on mixture parameters. The protocol grid is kept *uniform* at all times: after every daily update the node times are $\{0, 1/L, 2/L, \ldots, 1\}$. This is achieved by **compressing** the old protocol from $[0, 1]$ to $[0, L/(L+1)]$, **adding** the new day's density on the newly opened interval $[L/(L+1), 1]$, and **smoothing** the resulting $L + 1$ segments back to the fixed $L$-segment grid. An SDE realization of the same density path is discussed later, in Section 9.2 and Appendix A; it is not used by the CAS update.

Fig. 1 illustrates the recursion for $L = 4$.

**Compress–Add–Smooth recursion (illustrated for $L=4$)**

Figure 1: One iteration of the compress–add–smooth recursion, illustrated for $L = 4$ segments. **Top:** the protocol at day $n$ consists of $L$ uniform segments on $[0, 1]$. **Middle:** compression rescales the protocol to $[0, L/(L+1)]$; the new day is appended on $[L/(L+1)]$, producing $L+1$ uniform segments. **Bottom:** re-binning averages the $L+1$ segments back onto the $L$-segment grid. The right-hand labels indicate the information-theoretic role of each step: only smoothing is lossy. Dashed lines track a past-day readout time $t_{m|n}$, which contracts by factor $L/(L+1)$ every day.

### 2.1    Memory representation

At day $n$, the agent's memory consists of three objects:

(i)   a *prior distribution* $q^{(0)} \in \mathcal{G}_K$, where $\mathcal{G}_K$ is the class of $K$-component Gaussian mixtures;

(ii)  a *protocol grid*: $L + 1$ Gaussian-mixture states $\{G_j^{(n)}\}_{j=0}^{L}$, one at each node time $t_j = j/L$. Each $G_j^{(n)} \in \mathcal{G}_K$ is specified by weights $\pi_k^{(j)}$, means $m_k^{(j)} \in \mathbb{R}^d$, and covariances $\Sigma_k^{(j)} \in \mathbb{R}^{d \times d}$, $k = 1, \ldots, K$. Between adjacent nodes the density is defined by piecewise-linear interpolation of the GM parameters (see below);

(iii) a readout convention: a memory of age $a = n - m$ is queried at $t(a) = (L/(L+1))^a$. This formula need not be stored as an $O(n)$ dictionary in a deployed memory; the dictionary used in our experiments is an offline diagnostic convenience.

The production memory cost is $O(LKd^2)$ for the protocol grid ($L+1$ nodes, each storing $K$ weights, means of size $d$, and covariance matrices of size $d^2$) plus $O(Kd^2)$ for the prior. The stored memory is independent of the number of days already processed.

**Piecewise-linear interpolation.**   For any $t \in [t_j, t_{j+1}]$, with $\alpha = (t-t_j)/(t_{j+1}-t_j)$, the marginal density is the Gaussian mixture with linearly interpolated parameters:

$$\pi_k(t) = (1-\alpha)\,\pi_k^{(j)} + \alpha\,\pi_k^{(j+1)}, \quad m_k(t) = (1-\alpha)\,m_k^{(j)} + \alpha\,m_k^{(j+1)}, \quad \Sigma_k(t) = (1-\alpha)\,\Sigma_k^{(j)} + \alpha\,\Sigma_k^{(j+1)}. \quad (1)$$

This interpolation preserves the GM structure: for every $t$, the marginal is a valid $K$-component Gaussian mixture (weights sum to 1, covariances are positive definite by convexity). The interpolation is component-wise and therefore assumes labeled components. If daily mixtures are obtained by unlabeled fitting, a permutation-alignment step, for example Hungarian matching or entropic optimal transport between adjacent days, must be applied before interpolation. The deterministic CAS algorithm only needs the interpolated density path; an SDE drift can be reconstructed later from this path when sample trajectories are required.

### 2.2   Initialisation (day 1)

The initial (day 1) protocol is set up by linearly interpolating from the prior distribution $q^{(0)}$ at $t=0$ to the first day's target $q^{(1)}$ at $t=1$:

$$p_t^{(1)} = (1-t)q^{(0)} + tq^{(1)}, \tag{2}$$

where the linear combination acts on the GM parameters as in (1). The $L+1$ initial node states are obtained by evaluating this interpolant at the grid times $t_j = j/L$:

$$G_j^{(1)} = \left(1 - j/L\right)q^{(0)} + \left(j/L\right)q^{(1)}, \qquad j = 0, \dots, L. \tag{3}$$

(A nonlinear interpolant can also be used.) No drift reconstruction is needed for the CAS update; the density-to-SDE construction is deferred to Appendix A.

### 2.3   Step 1: exact compression

The old protocol, defined on $L+1$ nodes at times $\{0, 1/L, \dots, 1\}$, is mapped exactly to the sub-interval $[0, L/(L+1)]$ by re-labelling the node times:

$$G_j^{(n+1,\mathrm{cmp})} = G_j^{(n)}, \qquad \text{at time } t_j^{\mathrm{cmp}} = \frac{j}{L} \cdot \frac{L}{L+1} = \frac{j}{L+1}, \qquad j = 0, \dots, L. \tag{4}$$

The GM states at each node are unchanged; only the time labels are rescaled. This is an exact, lossless operation: the compressed protocol defines the same density path, played at $L/(L+1)$ speed.

### 2.4   Step 2: addition

A single new node is appended at $t=1$ with state $q^{(n+1)}$ (the new day's target distribution). The compressed grid already has a node at $t = L/(L+1)$ with state $G_L^{(n)}$ (the previous day's terminal marginal). Between these two nodes the density is again defined by linear interpolation:

$$p_t^{(n+1)} = \frac{1-t}{1 - L/(L+1)}\,G_L^{(n)} + \frac{t - L/(L+1)}{1 - L/(L+1)}\,q^{(n+1)}, \qquad t \in \left[L/(L+1),\, 1\right]. \tag{5}$$

After addition, the augmented protocol has $L+2$ nodes at the uniform grid $\{0, \frac{1}{L+1}, \frac{2}{L+1}, \dots, \frac{L}{L+1}, 1\}$, constituting $L+1$ segments of width $1/(L+1)$.

### 2.5   Step 3: smoothing by uniform-grid rebinning

The augmented protocol has $L + 2$ nodes (constituting $L + 1$ segments), but the budget allows only $L$ segments ($L + 1$ nodes). We restore the budget by *re-binning*: evaluating the augmented piecewise-linear density interpolant at the target grid and storing the resulting GM states as the new nodes.

Concretely, the augmented grid has nodes at $t_k^{\text{aug}} = k/(L + 1)$, $k = 0, \dots, L + 1$, and the target grid has nodes at $t_j^{\text{new}} = j/L$, $j = 0, \dots, L$. For each target node, we evaluate the augmented interpolant:

$$G_j^{\text{new}} = p_{t_j^{\text{new}}}^{\text{aug}}, \qquad j = 0, \dots, L. \tag{6}$$

Since $t_j^{\text{new}}$ falls inside some augmented segment $[t_k^{\text{aug}}, t_{k+1}^{\text{aug}}]$, the evaluation is a linear interpolation between two adjacent augmented nodes:

$$G_j^{\text{new}} = (1 - \alpha_{jk})\, G_k^{\text{aug}} + \alpha_{jk}\, G_{k+1}^{\text{aug}}, \qquad \alpha_{jk} = \frac{t_j^{\text{new}} - t_k^{\text{aug}}}{t_{k+1}^{\text{aug}} - t_k^{\text{aug}}}, \tag{7}$$

where $k$ is the unique index such that $t_k^{\text{aug}} \le t_j^{\text{new}} < t_{k+1}^{\text{aug}}$. Since the interpolation acts componentwise on the GM parameters $(\pi, m, \Sigma)$, the result is a valid $K$-component Gaussian mixture at every node (weights are convex combinations summing to 1; covariances remain positive definite by convexity of the PSD cone).

Equivalently, the operation can be written as a matrix–vector product using a sparse *re-binning matrix* $W \in \mathbb{R}^{(L+1)\times(L+2)}$ whose rows encode the interpolation weights (7). Each row of $W$ has at most two nonzero entries and sums to 1. The matrix depends only on $L$ and is precomputed once.

The entire smoothing step requires $O(LKd^2)$ operations: for each of $L + 1$ target nodes, interpolate the $K \times (d^2 + d + 1)$ GM parameters. No optimiser, no merge-pair selection, and no policy choice is needed.

### 2.6   Readout-time evolution

For analysis it is useful to track the query time associated with day $m$ at current day $n$:

$$t_{m|n} = \left(\frac{L}{L + 1}\right)^{n-m}. \tag{8}$$

Equivalently, if one explicitly tracks readout times during an offline experiment, they obey

$$t_{m|n+1} = \frac{L}{L + 1} t_{m|n}, \qquad t_{n+1|n+1} = 1. \tag{9}$$

This explicit dictionary is not part of the fixed production memory: a deployed system can compute the query time from the requested age $a = n - m$ using (8). The smoothing step does not move readout times; it changes the protocol node states, so the marginal evaluated at a fixed age-dependent readout time changes. This is the forgetting mechanism. For example, with $L = 10$, a 20-day-old memory is queried at $(10/11)^{20} \approx 0.12$, placing it in the leftmost 12% of the replay interval.

### 2.7   Computational cost

**Per-day update:** $O(LKd^2)$. Compression relabels $L + 1$ node times ($O(1)$ per node; the GM states are unchanged). Addition appends one new node and evaluates one interpolation ($O(Kd^2)$). Smoothing evaluates the augmented interpolant at $L + 1$ target nodes ($O(Kd^2)$ per node), giving $O(LKd^2)$ total. No backpropagation, no sampling, no optimiser.

**Per-replay query:** $O(Kd^2)$. The replay marginal at readout time $t_{m|n}$ is obtained by evaluating the piecewise-linear interpolant (1): locate the enclosing segment ($O(\log L)$ or $O(1)$ with a uniform grid), then interpolate $K$ means ($Kd$ operations) and $K$ covariance matrices ($Kd^2$ operations).

**Memory footprint:** For $d = 8$, $K = 3$, $L = 20$: the protocol occupies $21 \times 3 \times (64 + 8 + 1) = 4599$ floats $\approx 37\,\text{kB}$ in double precision, plus $\sim 1.8\,\text{kB}$ for the prior. No stored raw data, no replay buffer, and no $O(n)$ readout dictionary are required in the production representation. The original daily targets stored in our code are used only to compute retrospective evaluation metrics.

## 3 Forgetting-by-compression

In standard continual learning, forgetting arises from *parameter interference* McCloskey and Cohen (1989); French (1999): gradient updates on new data overwrite representations needed for old tasks. In the density-compression setting studied here, forgetting has a different and explicitly localized origin. The three steps have distinct mechanistic roles:

- *Compression* is **lossless** — it is an exact time-rescaling that preserves the marginal flow.
- *Addition* is **non-destructive** — the new day occupies a separate interval $[L/(L+1), 1]$ and does not modify the old protocol on $[0, L/(L+1)]$.
- *Smoothing* is **lossy** — rebinning replaces a finer grid by a coarser one, erasing sub-grid temporal detail.

Forgetting is therefore *localised in a single identifiable step*: the re-approximation of an $(L+1)$-segment protocol by an $L$-segment protocol via interpolant evaluation on a coarser grid. The temporal resolution available for old memories shrinks geometrically with age through the readout-time decay (8), making forgetting a consequence of *temporal coarse-graining* rather than parametric interference.

**Remark 1** (Temporal blurring of node states)**.** *Each re-binning cycle replaces node states by convex combinations of their neighbors, progressively smoothing the spatial variation along the protocol. Older (leftward) nodes have undergone more re-binning cycles and their GM parameters are therefore more blurred — component means are pulled toward a common average, covariances are inflated, and weight contrasts are reduced. This cumulative blurring is the microscopic mechanism behind the macroscopic forgetting curve. It can serve as a diagnostic: when leftward nodes become nearly indistinguishable, the memory is close to saturation.*

## 4 Forgetting metrics

We use moment-based metrics as the primary forgetting diagnostics throughout this paper. They are cheap to evaluate, analytically transparent, and sufficient for the GM class. For richer density families (e.g. neural parameterisations), distributional metrics such as KL divergence or Wasserstein-2 distance would be natural alternatives; we leave their systematic study to future work.

### 4.1 Raw moment mismatch

The replay distribution of past day $m$ at current day $n$ is $\widehat{p}^{(m|n)} = p_{t_{m|n}}^{(n)}$, the marginal of the current protocol evaluated at the readout time via (1). The raw forgetting metric is

$$F_{m \to n} \;=\; \|\mu_{\text{replay}} - \mu_{\text{orig}}\|^2 + \|\Sigma_{\text{replay}} - \Sigma_{\text{orig}}\|_F^2, \tag{10}$$

where $(\mu, \Sigma)$ are the overall mean and covariance of the Gaussian mixture, computed analytically from the GM parameters.

Rather than studying the full $(m, n)$ matrix, we work primarily with the *age variable* $a = n - m$ and define the age-dependent forgetting curve

$$\bar{F}(a) \;=\; \big\langle \bar{F}_{m \to n} \big\rangle_{n - m = a} \tag{11}$$

as the average over all pairs with $n - m = a$.

### 4.2 Normalized metric

To compare across $K$, $d$, and geometric scale, we normalize by the *amnesia baseline*:

$$F_{\text{amnesia}}(m) = \|\mu_0 - \mu_{\text{orig}}^{(m)}\|^2 + \|\Sigma_0 - \Sigma_{\text{orig}}^{(m)}\|_F^2, \tag{12}$$

where $(\mu_0, \Sigma_0)$ are the moments of the starting distribution (for a deterministic start: $\mu_0 = x_0$, $\Sigma_0 = 0$). The normalized forgetting is

$$\bar{F}_{m \to n} = \frac{F_{m \to n}}{F_{\text{amnesia}}(m)} \in [0, \infty). \tag{13}$$

Here $\bar{F} = 0$ is perfect recall, $\bar{F} = 1$ is total amnesia, and $\bar{F} > 1$ indicates *confusion* — the replay is actively worse than having no memory at all. The retention half-life is $a_{1/2} = \min\{a : \bar{F}(a) \geq \theta\}$ with threshold $\theta = 0.5$.

**Remark 2** (Confusion: $\bar{F} > 1$)**.** *When $\bar{F} > 1$, old memories have been pulled toward the current day's location rather than decaying toward the prior. We call this regime* confusion *to distinguish it from* destruction *($\bar{F} \to 1$, reversion to the uninformed prior).*

### 4.3 Decomposed metric for Gaussian mixtures

For $K > 1$, we decompose forgetting into per-component contributions after Hungarian matching:

$$F = F_{\text{mean}} + F_{\text{cov}} + F_{\text{weight}},$$
$$F_{\text{mean}} = \sum_k \bar{w}_k \|\Delta m_k\|^2, \quad F_{\text{cov}} = \sum_k \bar{w}_k \|\Delta \Sigma_k\|_F^2, \quad F_{\text{weight}} = \|\Delta \pi\|^2, \tag{14}$$

where $\bar{w}_k = \max(\pi_k^{\text{replay}}, \pi_k^{\text{orig}})$ and matching is by pairwise mean distance.

### 4.4 Distribution-sensitive diagnostics

The moment metric (10) is useful because it is analytic and inexpensive, but it deliberately compresses each mixture to its global mean and covariance. We therefore also use two density-level diagnostics to test whether the observed forgetting behavior survives beyond this reduction. Both are permutation-invariant with respect to mixture labels because they operate on either samples from the full density or likelihood under the full density.

First, we estimate a normalized sliced-Wasserstein distortion. For each original/replayed pair, we draw samples $X_i \sim p^{(m)}$ and $Y_i \sim \hat{p}^{(m|n)}$, project them onto random unit directions $\nu_\ell$, compute one-dimensional empirical Wasserstein distances, and average over directions. We normalize by the corresponding prior-vs-original amnesia baseline. Second, we compute a held-out negative-log-likelihood distortion,

$$D_{\text{NLL}}(m, n) = \mathbb{E}_{X \sim p^{(m)}} \left[ -\log \hat{p}^{(m|n)}(X) + \log p^{(m)}(X) \right], \tag{15}$$

again normalized by the prior-vs-original baseline. In practice these expectations are estimated on fixed held-out samples from each daily target distribution. Because these diagnostics are sample-based and more expensive than moment mismatch, we report final-day curves $n = N$ as a lightweight stress test, while the original moment curves remain age-averaged over all $(m, n)$ pairs.

## 5 Experiments: single-Gaussian ($K = 1$, $d = 2$)

### 5.1 Setup

We consider a stream of $n = 100$ daily Gaussian targets

$$p^{(m)} = \mathcal{N}\big(\mu^{(m)}, \Sigma\big), \qquad m = 1, \ldots, n,$$

in dimension $d = 2$, with fixed covariance

$$\Sigma = 0.5\, I.$$

Unless stated otherwise, the daily means follow a circular drift of radius $R = 2$,

$$\mu^{(m)} = R\big(\cos(2\pi m/P),\, \sin(2\pi m/P)\big), \qquad P = 50,$$

so that over the 100-day horizon the mean completes two full revolutions. The prior is $q^{(0)} = \mathcal{N}(0, I)$.

The default segment budget is

$$L = 10.$$

The circular drift is a deliberately nontrivial geometry. It is simple enough to visualize, but unlike a monotone linear drift it periodically revisits earlier spatial locations. This makes it possible to separate two effects: genuine temporal forgetting and geometric aliasing caused by revisiting the same region of state space at different times. To assess the role of geometry, we also compare against a linear-drift experiment in which the daily means move along a line at comparable local speed.

## 5.2 Default behavior: age curve, heatmap, and replay geometry

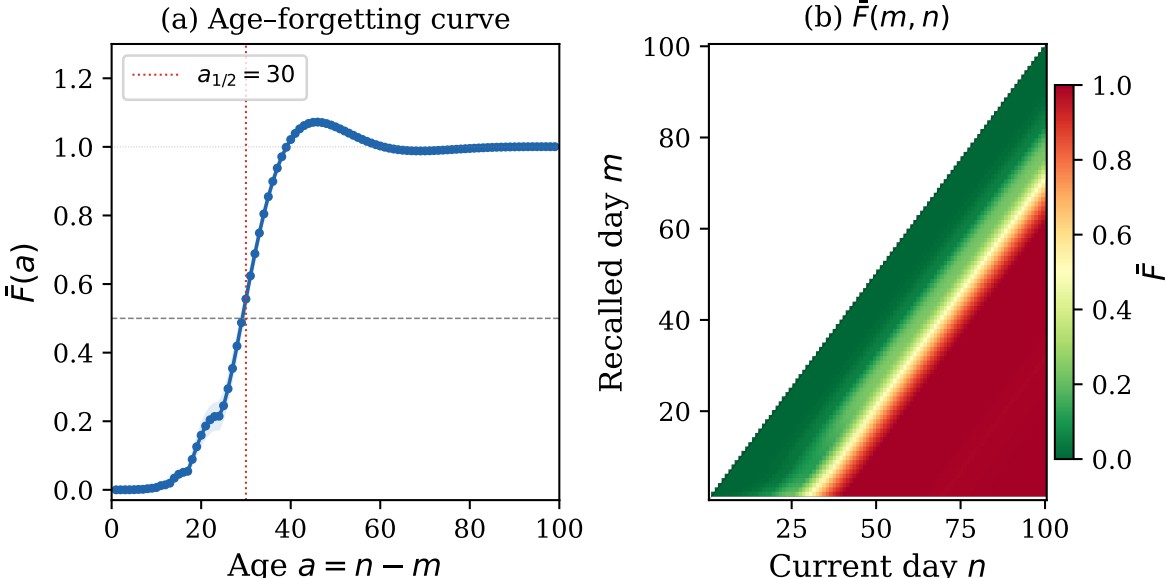

Figure 2: Single-Gaussian experiment ($K = 1$, $d = 2$, $L = 10$) under the default circular-drift setting. (a) Age-averaged normalized forgetting $\bar{F}(a)$, showing retention half-life $a_{1/2} = 30$. The curve exhibits a low-error plateau for ages 0–15, followed by a steep sigmoid transition crossing $\bar{F} = 0.5$ at age 30, a slight overshoot to $\bar{F} \approx 1.08$ around age 50 (the confusion regime), and eventual saturation near $\bar{F} = 1.0$. The curve is weakly non-monotone due to the periodic geometry of the circular drift, which causes geometric recurrence at multiples of the half-period. (b) Full forgetting matrix $\bar{F}(m, n)$ as a function of recalled day $m$ and current day $n$. The dominant trend is age-controlled forgetting (iso-forgetting contours run parallel to the diagonal), modulated by the periodic geometry of the underlying drift.

Fig. 2 shows the basic forgetting diagnostics for the default parameters. Panel (a) reveals a characteristic two-regime structure: recent memories ($a \lesssim 15$) are recalled with near-zero error, while older memories undergo a rapid sigmoid-like degradation. The half-life $a_{1/2} = 30$ means that, with $L = 10$ segments, the agent retains useful recall of the past ~30 days. The slight overshoot $\bar{F} > 1$ in the age range 40–60 confirms the confusion phenomenon (Remark 2): old replayed means are pulled toward the current day's location rather than decaying to the prior. Panel (b) shows that the dominant structure of the forgetting matrix is age-controlled, with periodic modulation visible as faint stripes at multiples of the half-period $P/2 = 25$.

Fig. 3(a) visualizes the replayed means at the final day $n = 100$. Recent memories are replayed close to their true locations on the right-lower arc of the circle, whereas older memories are displaced toward a compressed cluster near the origin. This spatial collapse is the geometric signature of confusion: old replayed means are attracted toward the time-weighted average of the protocol, which is dominated by recent days.

Fig. 4 makes the confusion mechanism visible: as age increases, the replayed mean migrates inward from the true location on the circle toward the origin (the time-averaged protocol centre), while the replayed covariance

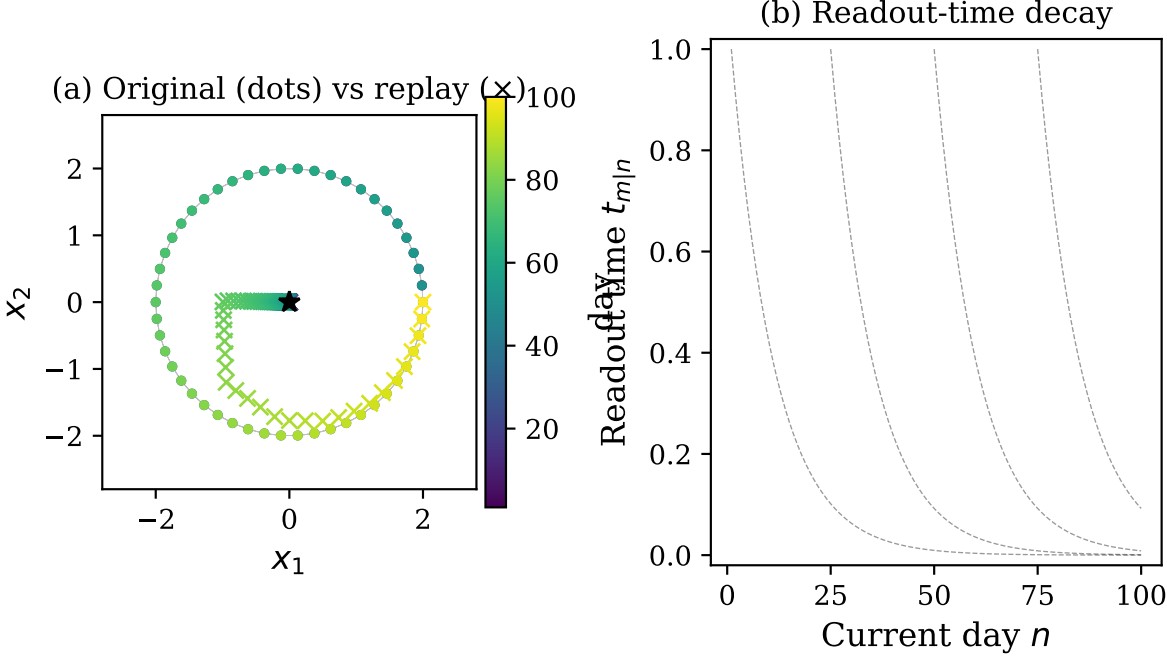

Figure 3: Default single-Gaussian circular-drift experiment ($L = 10$). (a) Original daily means (dots, coloured by day) and replayed means at the final day (crosses). The black star marks the prior mean (the origin), which serves as the protocol's long-time attractor. Confusion is visible as the systematic inward displacement of crosses from the circle toward the star: recent memories (warm colours) are replayed near their true locations on the circle, while older memories (cool colours) are pulled progressively toward the prior mean. This convergence of replay means toward the star — rather than remaining on the circle — is the geometric signature of confusion. (b) Readout times $t_{m|n}$ versus current day $n$, showing the geometric decay (8).

inflates dramatically. The arrows connecting original to replayed positions show that the displacement is systematically directed toward the protocol interior, not random.

Fig. 3(b) shows the readout times $t_{m|n}$ versus current day $n$. They decay geometrically as $(L/(L+1))^{n-m}$, with the theoretical curves (dashed) overlaid for reference. The actual and theoretical curves coincide exactly, confirming the readout-time evolution (8). For $L = 10$, a 30-day-old memory sits at $t \approx (10/11)^{30} \approx 0.047$, deep in the leftward portion of the protocol where rebinning-induced blurring is most severe.

### 5.3 Parameter dependence

The segment budget $L$ is the primary determinant of retention. Sweeping $L \in \{5, 8, 10, 15, 20, 30\}$ yields half-lives $a_{1/2} \in \{14, 24, 30, 44, 51, 74\}$, scaling roughly as $a_{1/2} \approx 2.4\,L$ (Fig. 5). This near-linear scaling is consistent with the observation that each CAS cycle degrades the readout time by a factor $L/(L+1)$, so the number of cycles before a memory reaches a fixed resolution threshold is proportional to $L$.

Drift speed modulates the half-life: faster drift (shorter period $P$) leads to shorter retention, because larger daily displacements accumulate more error through rebinning. Sweeping $P \in \{25, 50, 100, 200\}$ yields $a_{1/2} \in \{20, 30, 34, 36\}$. The dependence saturates for slow drift ($P \geq 100$), suggesting a floor set by the diffusive contribution of the rebinning itself.

Drift geometry (circular vs. linear) affects the curve shape more than the half-life: linear drift yields a clean monotone sigmoid with $a_{1/2} = 42$ (vs. 30 for circular at the same $L$), while circular drift introduces non-monotone modulations due to periodic spatial recurrence. The higher linear-drift half-life reflects the absence of geometric aliasing: each recalled location is unique, so the rebinning error is always genuine.

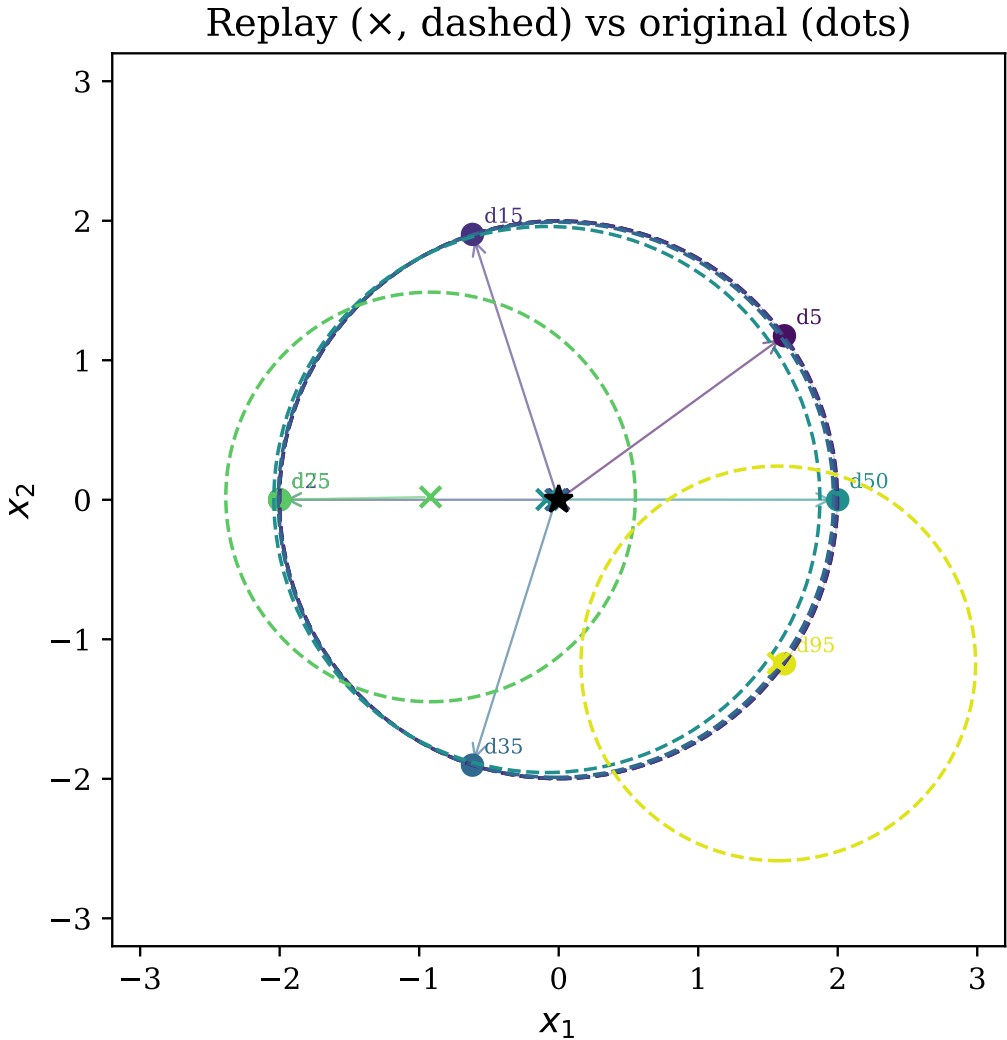

Figure 4: Selected replay ellipses in the default single-Gaussian circular-drift experiment ($L = 10$). For each displayed day, the original target mean is shown by a dot, while the replayed mean is shown by a cross with a dashed ellipse representing the replay covariance. Recent memories (e.g. day 95) are replayed with small displacement and compact ellipses. Intermediate-age memories (e.g. day 35) show large displacement toward the origin and inflated covariances. Very old memories (e.g. day 25) collapse nearly to the origin with very large ellipses. Geometric aliasing is visible for day 5, whose true location on the circle lies close to day 95 (they differ by two full periods), producing an apparently accurate replay that is coincidental rather than genuine recall.

### 5.4 Takeaway

The $K = 1$ experiments establish three main observations under the present moment-based diagnostic. First, the forgetting curve has a reproducible two-regime shape (plateau + sigmoid), and the transition shifts approximately linearly with the segment budget $L$ in this smooth circular-drift source. Second, the proportionality constant is source-dependent: drift speed changes the half-life, and drift geometry changes both the curve shape and the apparent transition age. Third, forgetting manifests as confusion (displacement toward recent eras), not destruction (reversion to the prior). These observations motivate the $K > 1$ experiments below, where we test whether this temporal-compression picture persists for labeled mixtures.

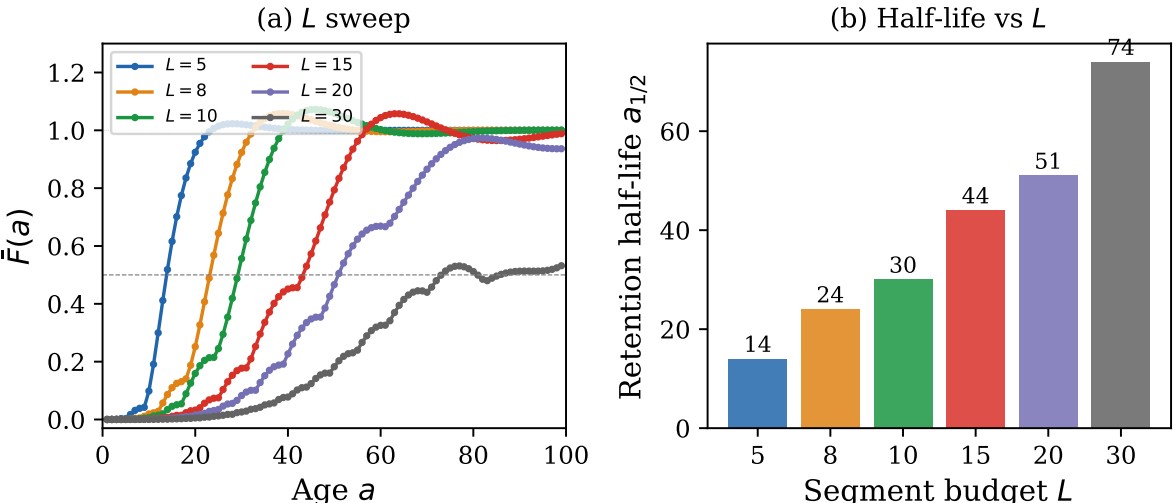

Figure 5: Segment-budget sweep for the single-Gaussian circular-drift experiment. (a) Age–forgetting curves for $L \in \{5, 8, 10, 15, 20, 30\}$. Increasing $L$ shifts the sigmoid transition to higher ages without changing the curve shape qualitatively. (b) Retention half-life $a_{1/2}$ versus $L$, confirming approximate linear scaling.

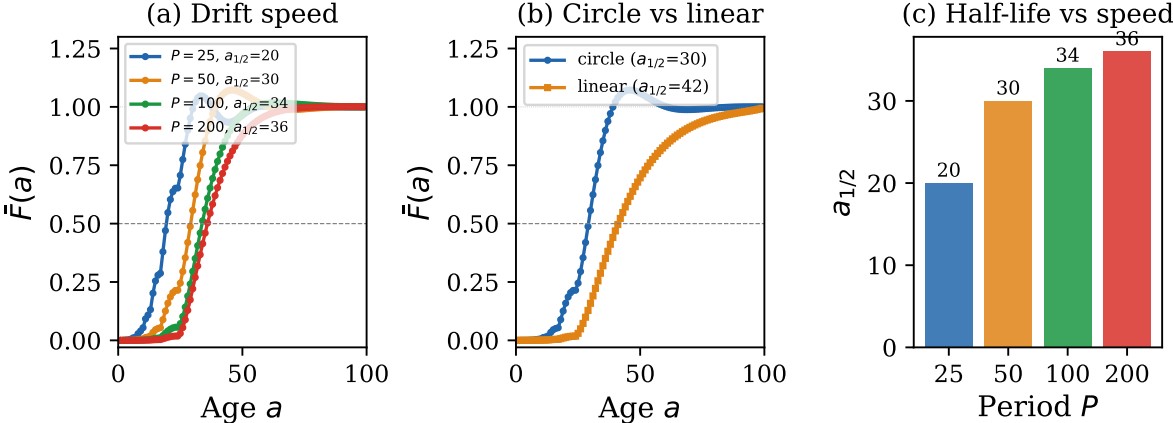

Figure 6: (a) Drift-speed sweep (circle period $P$). Faster drift shortens the half-life, but the effect saturates for slow drift. (b) Circle vs. linear drift. The circular curve is non-monotone due to periodic spatial recurrence; the linear curve is a clean sigmoid with longer half-life. (c) Half-life versus period $P$.

# 6 Experiments: Gaussian mixtures ($K = 3$, $d = 2$)

We now extend to $K$-component Gaussian-mixture daily targets. Each day's distribution has $K = 3$ equal-weight components arranged in a rotating equilateral triangle of radius $r = 0.8$ around the drifting circle centre (same circular drift as Section 5, with per-component covariance $0.3\,I$).

## 6.1 Default run and decomposed forgetting

With $L = 10$, the $K = 3$ experiment yields $a_{1/2} = 30$ — *identical* to the $K = 1$ case (Fig. 7a). This is the first indication that retention is governed by the temporal budget $L$ rather than the state-space complexity $K$.

The decomposed forgetting (Fig. 7b) reveals that $F_{\mathrm{mean}}$ dominates ($\sim$85% of total raw forgetting), $F_{\mathrm{cov}}$ contributes $\sim$15%, and $F_{\mathrm{weight}}$ is negligible (of order $10^{-17}$, i.e. machine precision). The vanishing weight

error is a structural consequence of equal-weight mixtures: convex combinations of equal weights remain equal, so the rebinning preserves weights exactly.

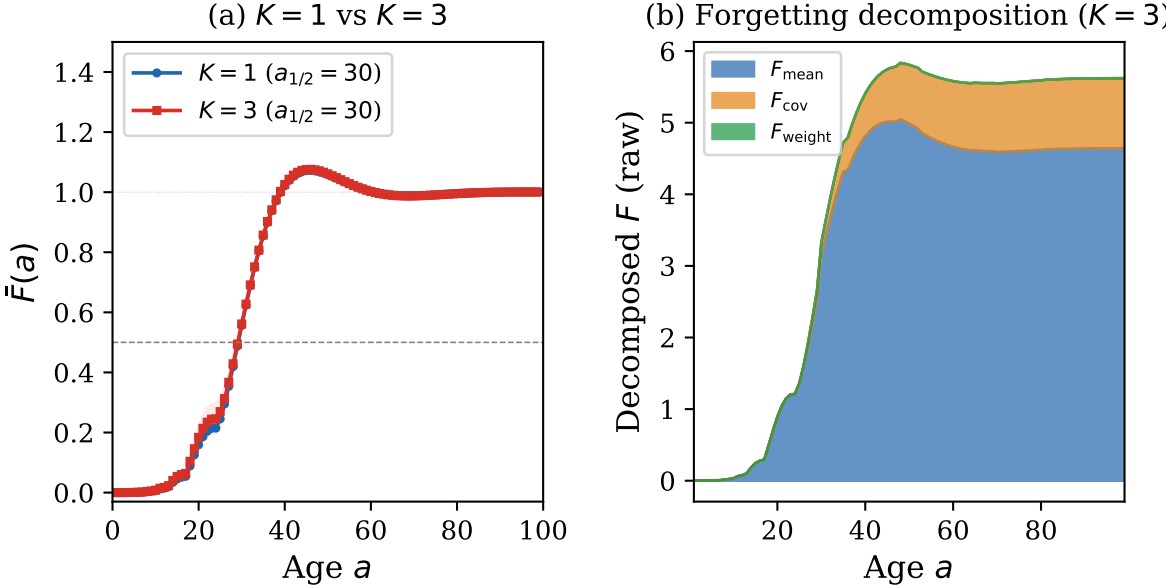

Figure 7: (a) Age–forgetting curves for $K = 1$ (blue) and $K = 3$ (red), both with $L = 10$ and circular drift. The curves nearly coincide, with both yielding $a_{1/2} = 30$. (b) Decomposed raw forgetting for $K = 3$: mean misalignment (blue) dominates, covariance error (orange) is secondary, and weight error (green) is negligible.

## 6.2 Component-level trajectories

Fig. 8 shows the per-component replayed means (after Hungarian matching) at the final day. Recent days' component means are replayed accurately; older days collapse toward the protocol interior, with all three components converging toward a common cluster near the origin. This mirrors the $K = 1$ confusion pattern, amplified by the need to simultaneously track three interacting trajectories.

## 6.3 $L$ sweep and $K$ sweep

Sweeping the segment budget at $K = 3$ gives half-lives $a_{1/2} \in \{14, 30, 41, 50, 71\}$ for $L \in \{5, 10, 15, 20, 30\}$ (Fig. 9a), closely matching the $K = 1$ results.

The $K$ sweep at $L = 10$ (Fig. 9b) yields $a_{1/2} \in \{30, 29, 30, 30, 30\}$ for $K \in \{1, 2, 3, 5, 8\}$ — the half-life is essentially *flat* across mixture complexity. This is the paper's central experimental finding: *retention is controlled by the temporal budget $L$, not by the state-space complexity $K$*. The $K$-independence is not approximate: the half-life varies by at most one day across a factor-of-8 range in $K$.

## 6.4 Takeaway

The $K > 1$ experiments establish two main results. First, the half-life is independent of $K$: adding mixture components does not shorten (or lengthen) retention. This is because the re-binning step treats all GM parameters (weights, means, covariances) uniformly — the interpolation does not "see" how many components there are. Second, forgetting is overwhelmingly driven by mean misalignment; covariance error is secondary and weight error is negligible for equal-weight mixtures. These findings justify using the half-life $a_{1/2}$ as a single scalar summary of retention quality, controlled by $L$ alone.

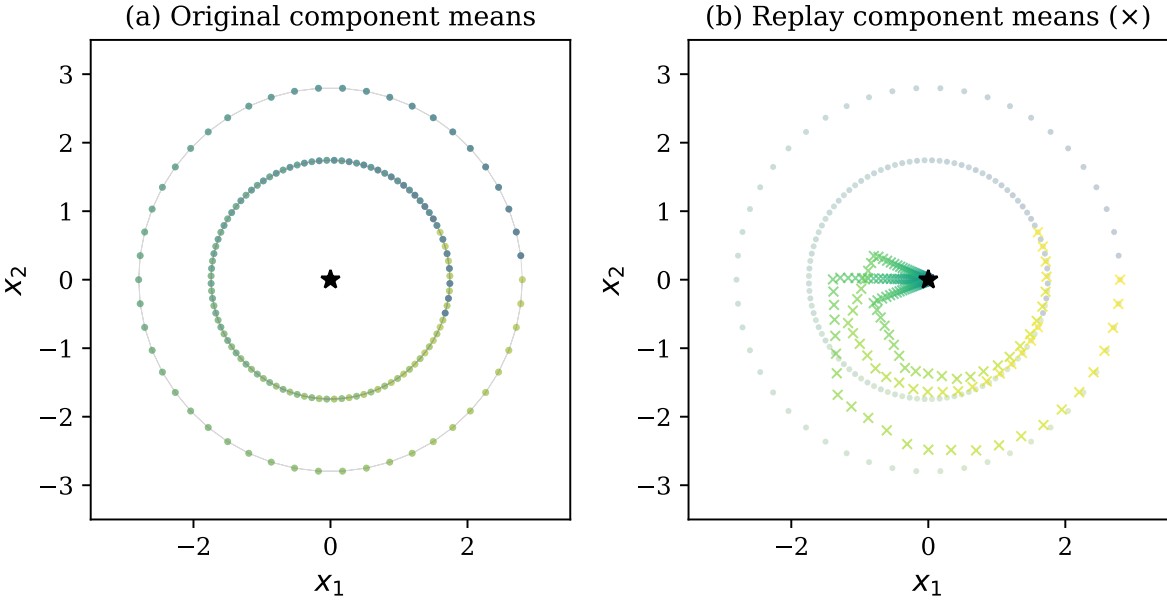

Figure 8: Component-level trajectories for $K = 3$. (a) Original daily component means, colored by day. The three interleaved helical paths trace the rotating-triangle geometry. (b) Replayed component means ($\times$) at the final day, after Hungarian matching to the originals. Recent components are well-recalled; older ones collapse toward the origin.

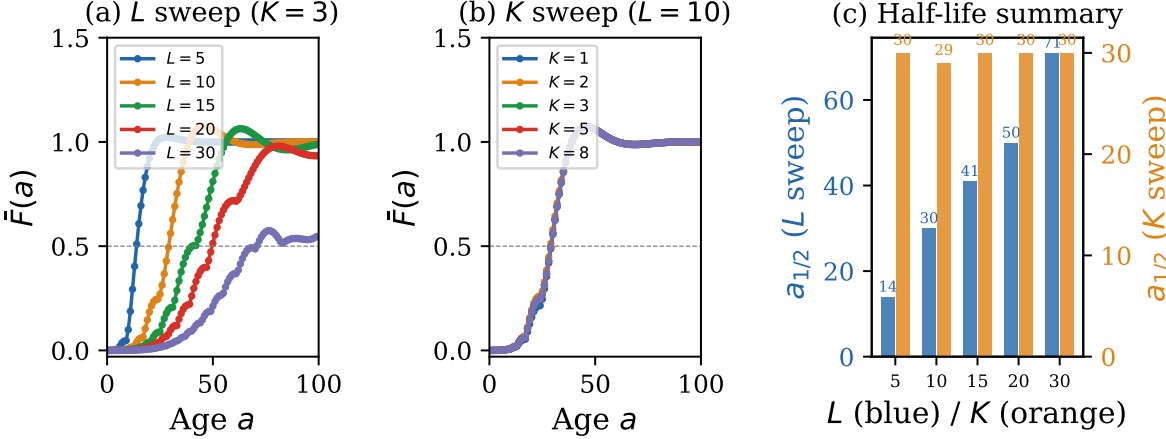

Figure 9: (a) $L$ sweep at $K = 3$: age–forgetting curves for $L \in \{5, 10, 15, 20, 30\}$. The pattern mirrors the $K = 1$ case. (b) $K$ sweep at $L = 10$: age curves for $K \in \{1, 2, 3, 5, 8\}$ nearly coincide. (c) Half-life summary for both sweeps.

## 6.5 Sliced-Wasserstein and held-out likelihood

We analyzed final-day density-level diagnostics to check whether the main retention picture survives beyond global moment mismatch. Fig. 10 compares the original moment metric with normalized sliced-Wasserstein and held-out NLL distortions for the default $K = 1$ and $K = 3$ streams. For $K = 1$, the final-day half-lives are 30 under the moment metric, 30 under sliced-Wasserstein, and 26 under held-out NLL. For $K = 3$, the corresponding values are 29, 29, and 21. Thus the density-level Wasserstein diagnostic is close to the moment metric in these smooth streams, whereas held-out likelihood is more sensitive and produces a shorter retention estimate.

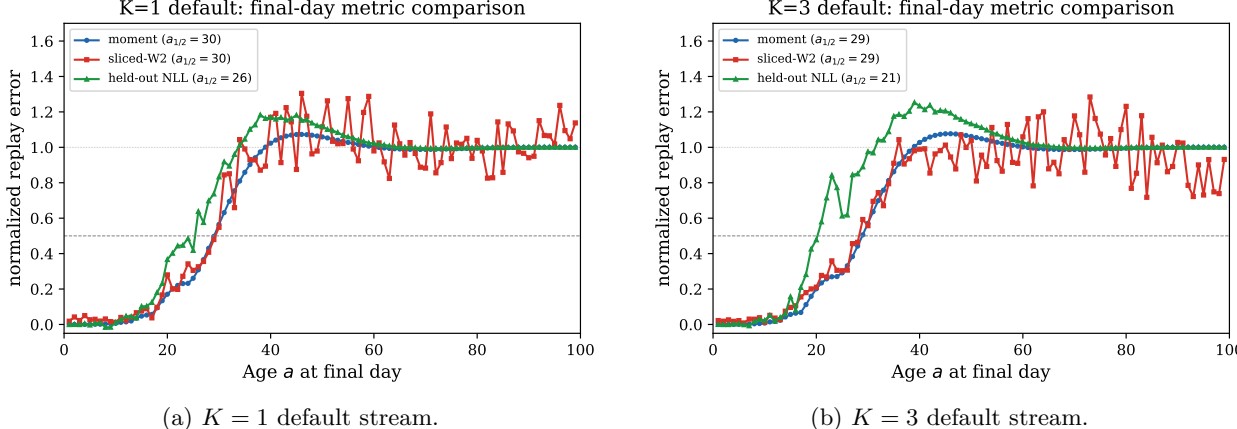

(a) $K = 1$ default stream.

(b) $K = 3$ default stream.

Figure 10: Diagnostics at the final day: normalized moment mismatch, sliced-Wasserstein, and held-out negative log likelihood. The reported half-life is metric-dependent.

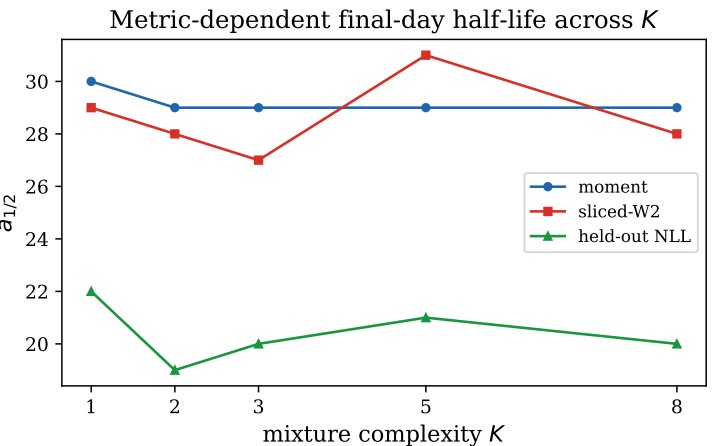

Figure 11: Final-day half-life across mixture complexity under three diagnostics. The conclusion is stated as metric-dependent: moment and sliced-Wasserstein diagnostics are nearly flat in $K$, while held-out likelihood is more stringent.

We also repeated the $K$-sweep at $L = 10$ using the final-day diagnostics (Fig. 11). The moment and sliced-Wasserstein half-lives remain nearly flat across $K \in \{1, 2, 3, 5, 8\}$, while held-out NLL gives consistently shorter half-lives, approximately 19–22 days. This supports the empirical observation that CAS is not strongly degraded by the number of labeled mixture components in this controlled experiment.

## 6.6 Memory-matched streaming baselines

Since CAS with segment budget $L = 10$ stores $L + 1 = 11$ Gaussian-mixture protocol nodes, all streaming baselines below are constrained to store the same number $B = 11$ of GM states. The prior is not counted in this comparison because it is common to all methods. We compare final-day normalized moment distortion, matching the lightweight stress-test protocol used in Section 6.5.

The baselines are deliberately simple. FIFO stores the most recent $B$ daily GM states and uses the oldest retained state for older queries. Reservoir stores the current state plus a uniform reservoir sample of $B - 1$ previous states; the plotted curve averages over five random seeds. Log-age is an online multi-resolution buffer that preserves nodes approximately uniformly in $\log(1 + \text{age})$. Greedy-PL is an online piecewise-linear compressor: after appending a new daily state, it deletes the interior knot whose removal gives the

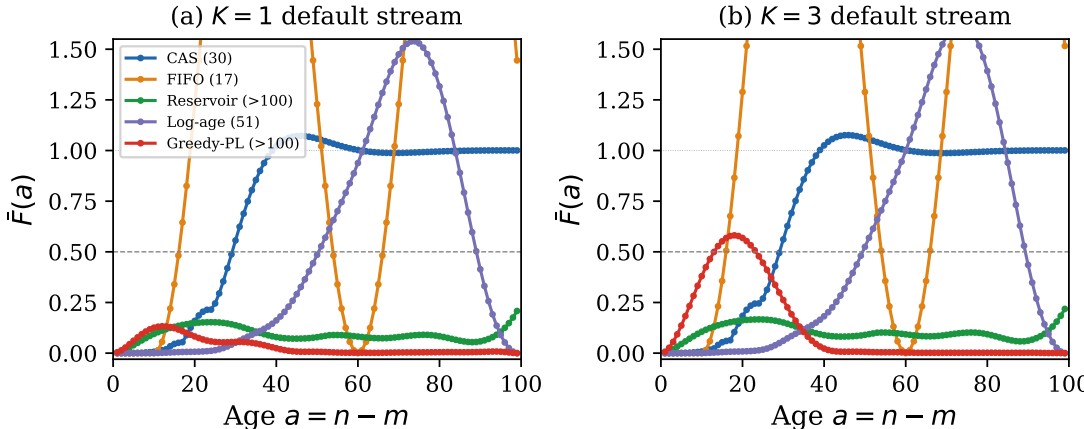

Figure 12: Memory-matched baseline comparison at the final day. CAS uses $L = 10$, i.e. 11 stored GM protocol states; every baseline stores the same $B = 11$ GM states. Labels report the first age at which the normalized moment error crosses 0.5; "> 100" means that the threshold was not crossed over the 100-day horizon. FIFO is weaker than CAS, while smoothness-aware temporal compression baselines can outperform CAS on these smooth streams.

smallest local moment-reconstruction error between its two neighboring knots. All baselines answer a query by interpolation, or by boundary extrapolation when the requested day lies outside the retained knots.

Fig. 12 shows the result for the default $K = 1$ and $K = 3$ streams. FIFO is weaker than CAS, with half-lives 17 and 16 days for $K = 1$ and $K = 3$, respectively, compared with CAS half-life 30 in both cases. Smoothness-aware baselines can be stronger on these very smooth circular streams: log-age gives half-lives 51 and 50, while reservoir and greedy piecewise-linear compression remain below the half-life threshold over the 100-day horizon. This comparison suggests the following interpretation: CAS substantially improves over a naive FIFO replay buffer, but it is not an optimal temporal compressor under the moment metric. The strong performance of log-age and greedy-PL also points to a natural extension of CAS: replacing the uniform protocol grid by an adaptive or multiresolution grid chosen by an explicit distortion criterion.

## 7 Scaling experiments

We now test how the continual memory mechanism scales when the daily targets become more crowded, when the relevant signal is embedded into a higher-dimensional ambient space, and when the target family undergoes a simple topological curriculum involving split-and-merge events. The goal of this section is not to optimize performance, but to identify which aspects of increasing problem complexity actually shorten retention and which do not.

Based on the $K$-independence result of Section 6 — specifically, the flat half-life across $K \in \{1, 2, 3, 5, 8\}$ — we conjectured that the same qualitative picture would persist: forgetting is governed primarily by temporal compression under a fixed protocol budget, while many forms of static state-space complexity affect the geometry of replay far more than the retention timescale itself. The experiments below confirm this conjecture under three increasingly challenging scenarios.

### 7.1 Crowding as a control parameter

We begin with mixtures in $d = 2$, varying the *crowding ratio* $\chi = r/\sigma$, where $r$ is the inter-component offset radius and $\sigma = \sqrt{\text{cov\_scale}}$ is the component standard deviation. Small $\chi$ corresponds to heavily overlapping components (strong crowding); large $\chi$ to well-separated components (weak crowding). We sweep $r \in \{0.15, 0.3, 0.5, 0.8, 1.2, 2.0\}$ at $K = 3$, corresponding to $\chi \in \{0.27, 0.55, 0.91, 1.46, 2.19, 3.65\}$. Fig. 13 summarizes the results.

Panel (a) shows retention half-life $a_{1/2}$ versus crowding ratio for $K = 2, 3, 5, 8$. The first observation is that all curves are *flat at $a_{1/2} = 30$ for $\chi \lesssim 1.5$*: moderate-to-strong crowding has no effect on retention whatsoever. Only at high separation ($\chi > 2$) does the half-life begin to decrease, dropping to $a_{1/2} \approx 20$ at $\chi = 3.65$. The effect is most pronounced for $K = 2$, whose half-life begins declining earlier (at $\chi \approx 1$) than for $K \geq 3$. This decline at large $\chi$ is a geometric effect: when components are widely separated, each component's mean displacement under rebinning is larger in absolute terms, accelerating forgetting.

Panel (b) shows the age–forgetting curves at $K = 3$ for six crowding values. The curves for $\chi \leq 1.5$ are nearly indistinguishable, all showing the standard sigmoid with $a_{1/2} = 30$. At $\chi = 2.2$ the half-life shortens slightly to 28, and at $\chi = 3.7$ it drops to 20, with the sigmoid onset shifting leftward and the confusion overshoot ($\bar{F} > 1$) increasing.

Panel (c) reports the average share of raw forgetting attributable to mean misalignment. The mean-error share is $\sim 90\%$ across all crowding ratios, confirming that even as crowding changes the spatial geometry of forgetting, the dominant error channel remains mean displacement rather than covariance distortion or weight drift.

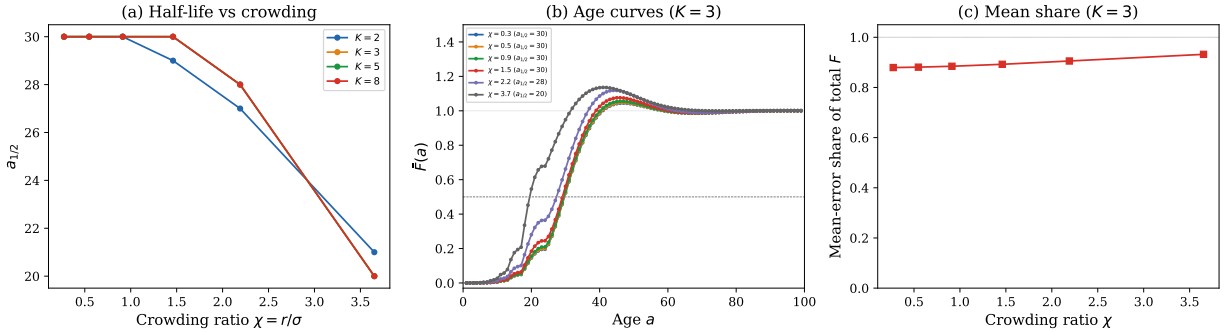

Figure 13: Crowding sweep for Gaussian-mixture memories ($L = 10$). (a) Retention half-life $a_{1/2}$ versus crowding ratio $\chi$ for $K = 2, 3, 5, 8$. The half-life is flat at 30 for $\chi \lesssim 1.5$ and declines only for well-separated components. (b) Age–forgetting curves at $K = 3$ for six crowding values. (c) Mean-error share of total forgetting, stable at $\sim 90\%$ across all $\chi$.

To illustrate the shape of forgetting more directly, Fig. 14 shows three representative age-forgetting curves corresponding to strong ($\chi = 0.3$), medium ($\chi = 0.9$), and weak ($\chi = 2.2$) crowding. The most notable feature is that the strong and medium crowding curves are virtually identical ($a_{1/2} = 30$ for both), while the weakly crowded case shows a slightly earlier sigmoid onset ($a_{1/2} = 28$) and a more pronounced confusion overshoot ($\bar{F} \approx 1.12$ vs. $\approx 1.05$). The overshoot amplification at weak crowding is consistent with larger per-component mean displacements when components are far apart.

## 7.2 Fixed low-dimensional signal in a higher-dimensional ambient space

We next test whether retention degrades when the informative signal remains two-dimensional but is embedded into a higher-dimensional ambient space. Fig. 15 reports the results for ambient dimensions $d = 2, 4, 8, 16$, with $K = 3$ and $L = 10$.

Panel (a) shows the age-forgetting curves when the extra dimensions carry no drift (nuisance coordinates remain at zero). The curves shift *rightward* with increasing $d$: the half-life increases slightly from $a_{1/2} = 30$ at $d = 2$ to $a_{1/2} = 34$ at $d = 16$. This counter-intuitive improvement occurs because the amnesia baseline $F_{\text{amnesia}}$ grows with $d$ (the prior covariance is $I_d$, contributing more Frobenius-norm distance from each daily target), while the rebinning error in the signal subspace is unchanged. The normalised forgetting is therefore diluted by the larger baseline.

Panel (b) summarizes the half-life as a function of $d$ for two settings. When nuisance dimensions are static, $a_{1/2}$ increases gently from 30 to 34. When nuisance dimensions carry a slow random walk (speed 0.1/day), the

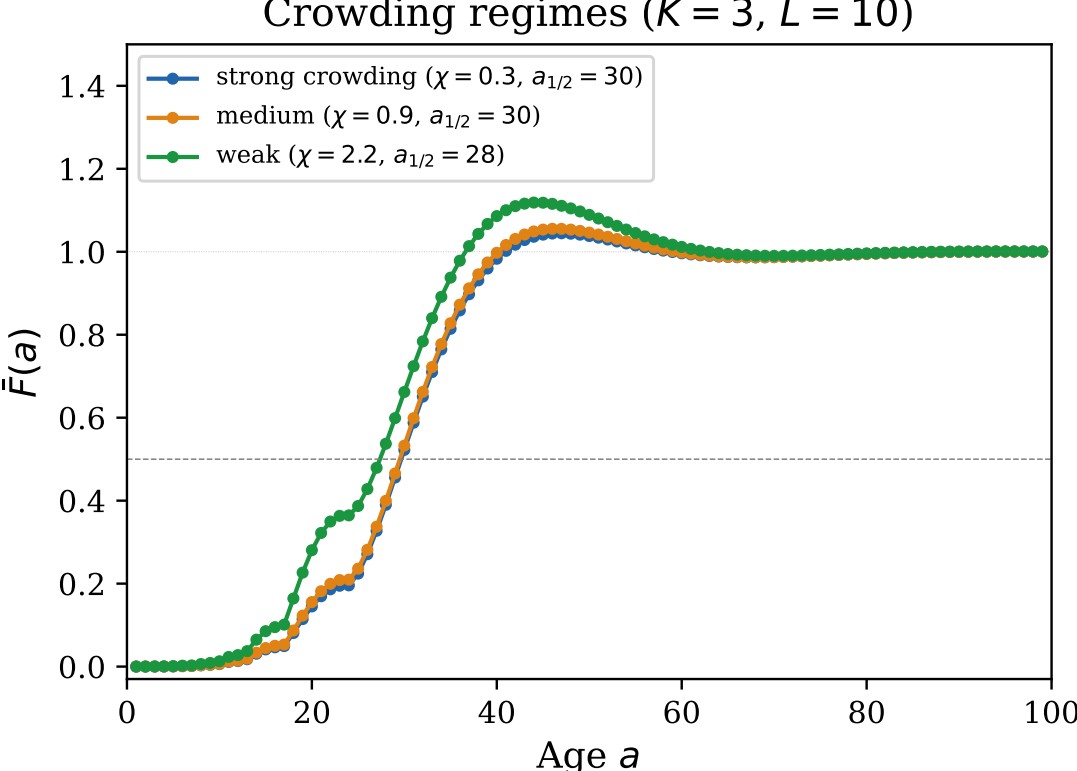

Figure 14: Representative age-averaged forgetting curves across three crowding regimes ($K = 3$, $L = 10$). Strong ($\chi = 0.3$) and medium ($\chi = 0.9$) crowding yield indistinguishable curves ($a_{1/2} = 30$). Weak crowding ($\chi = 2.2$) produces a slightly earlier transition ($a_{1/2} = 28$) and a more pronounced confusion overshoot.

half-life follows a similar trend ($a_{1/2} \approx 30$–$33$), indicating that moderate nuisance drift does not substantially impair retention of the signal.

Panel (c) shows that the mean-error share declines with $d$, from $\sim 90\%$ at $d = 2$ to $\sim 60\%$ at $d = 16$ in the no-nuisance setting. This shift reflects the growing contribution of covariance mismatch in the extra dimensions: as $d$ increases, the $d \times d$ covariance matrices carry more entries that can accumulate rebinning error. With nuisance drift, the mean-error share remains higher ($\sim 70\%$ at $d = 16$) because the drifting nuisance means contribute additional mean-channel error.

### 7.3 Split-and-merge curriculum

As a final scaling test, we consider a simple curriculum in which the daily mixture geometry changes topologically over time via split-and-merge events. The $K = 3$ mixture undergoes four phases, illustrated schematically in Fig. 16: a normal rotating triangle ($r = 0.8$, days 1–30), a merge phase where two components collapse toward each other ($r_{01} \to 0.05$, days 31–50), a split phase where they separate again ($r \to 0.8$, days 51–80), and a final collapse where all three components converge toward the centre ($r \to 0.1$, days 81–100). Transitions are smoothed over 5-day ramps. Fig. 17 shows both the daily component means and the resulting age-forgetting curve.

Panel (a) displays the component centres across the 100 days, with phase-boundary markers (red diamonds) at days 1, 31, 51, and 81. The four phases are clearly visible: the initial rotating triangle, the merged pair, the re-separation, and the final collapse.

Panel (b) shows the corresponding age-forgetting curve. Despite the nontrivial topological evolution, the half-life is $a_{1/2} = 30$ — identical to the stationary-geometry baseline. The curve shape is the standard sigmoid

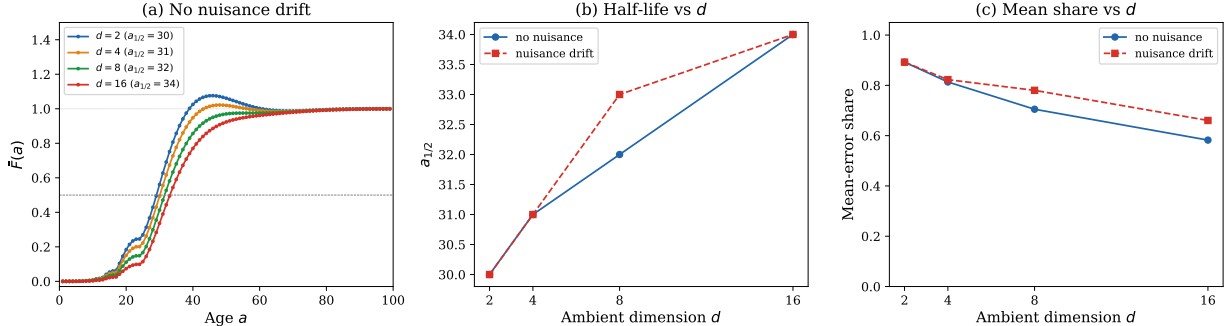

Figure 15: Scaling with ambient dimension for a fixed 2D signal ($K = 3$, $L = 10$). (a) Age-forgetting curves for $d = 2, 4, 8, 16$ without nuisance drift. Higher $d$ slightly *improves* normalised retention due to the larger amnesia baseline. (b) Retention half-life versus ambient dimension under two nuisance settings. Both show gentle increase with $d$. (c) Mean-error share decreases with $d$ as covariance mismatch grows in the extra dimensions.

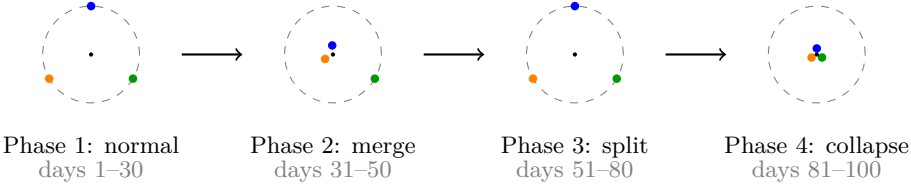

Figure 16: Schematic of the four-phase split-and-merge curriculum for $K = 3$. Coloured dots represent the three mixture component means; the dashed circle indicates the inter-component radius $r$. Phase 1: normal rotating triangle ($r = 0.8$). Phase 2: two components merge ($r_{01} \to 0.05$). Phase 3: split back to triangle ($r \to 0.8$). Phase 4: all three collapse toward the centre ($r \to 0.1$). Transitions are smoothed over 5-day ramps.

with a mild non-monotone feature around age 10–15, attributable to the interaction between curriculum transitions and the periodic drift geometry.

This result suggests that, for this labeled and smoothly varying curriculum under the moment metric, the observed retention timescale is dominated by the temporal budget $L$: even when the daily target distribution undergoes qualitative structural changes — merging, splitting, and collapsing of mixture components — the half-life is unchanged in this experiment.

### 7.4 Overview and interpretation

The three scaling experiments paint a consistent picture. Crowding affects the half-life only at extreme separation ($\chi > 2$, where per-component mean displacements become large), and even then the reduction is modest (from 30 to 20). Ambient dimension either has no effect or slightly *improves* normalized retention (due to the growing amnesia baseline), while shifting the forgetting channel from mean-dominated toward a more even mean/covariance split. A time-varying curriculum with topological changes leaves the half-life entirely unchanged.

Taken together, these results support the following conclusion: in the tested smooth Gaussian-mixture streams, and under the current moment-based normalization, the retention half-life is primarily controlled by the temporal budget $L$, while $K$, ambient dimension, crowding, and the split–merge curriculum have weaker effects than drift speed. The coefficient $a_{1/2} \approx cL$ is not universal. It is a property of the source, representation, and metric. Improving retention within the current framework can be attempted either by increasing $L$ or by replacing the uniform grid with an adaptive one that allocates temporal resolution according to an explicit distortion criterion.

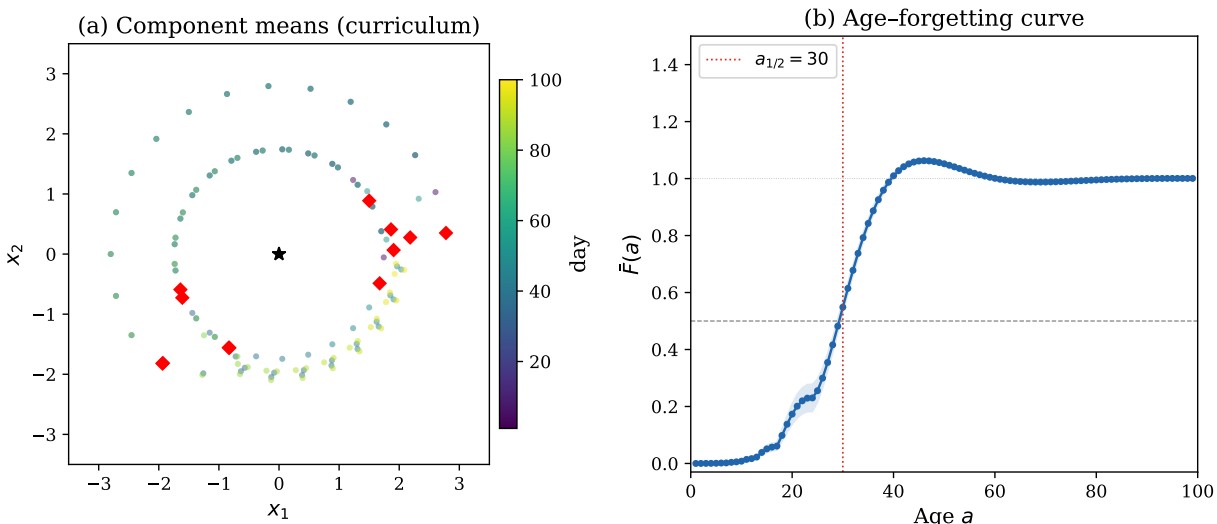

Figure 17: Split-and-merge curriculum experiment ($K = 3$, $L = 10$). (a) Daily component means, coloured by day, with phase-boundary markers (red diamonds) at days 1, 31, 51, and 81. The four phases — normal triangle, merge, split, collapse — are clearly visible. (b) Age-averaged forgetting curve. Despite the nontrivial curriculum, the retention half-life remains $a_{1/2} = 30$, identical to the stationary baseline.

## 8 MNIST latent-space illustration

To complement the controlled Gaussian-mixture experiments, we construct an image-based latent-space illustration using MNIST. The purpose is twofold: (i) to test whether age-dependent forgetting and retention-time control persist when the GM components represent real image classes; and (ii) to visualize the stored density protocol by decoding its nodes and interpolated marginals frame by frame. This visualization is a protocol-level marginal movie, not an experiment validating SDE trajectory replay.

### 8.1 Setup: latent embeddings and rotating-dominance curriculum

We select three visually distinct MNIST digit classes — 0, 3, and 8 — and embed the corresponding $\sim$18,000 training images into a $d = 12$ PCA latent space (57% explained variance). At this dimension, PCA-decoded class centroids are clearly recognizable as their respective digits (Fig. 18).

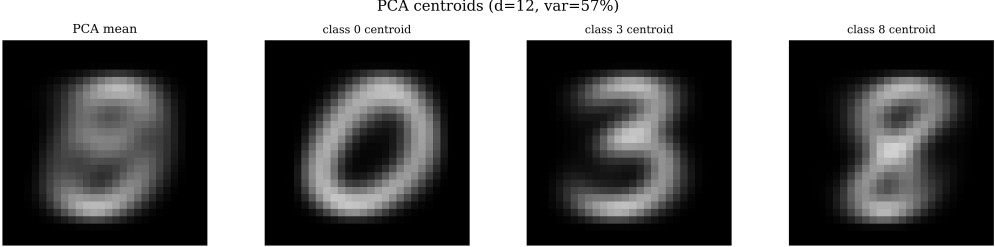

Figure 18: PCA centroids at $d = 12$: the global mean (left) and per-class centroids for digits 0, 3, and 8. Despite capturing only 57% of total variance, the centroids are visually recognisable.

We fit a single Gaussian per class ($K = 3$ total) and construct a *rotating-dominance curriculum* over $n = 100$ days: the component means and covariances are fixed to their class-conditional fits, while the mixing weights rotate with period $P = 30$:

$$\pi_k^{(m)} = \text{softmax}\big(A \cos(2\pi m/P + 2\pi k/3)\big), \qquad A = 2, \tag{16}$$

so that each digit class cycles between dominance ($\pi_k \approx 0.9$) and near-absence ($\pi_k \approx 0.04$). This is the semantic analogue of the synthetic circular drift: the "location" in distribution space rotates through digit classes rather than through spatial coordinates.

## 8.2 Forgetting curve and comparison with synthetic experiments

Running CAS with $L = 10$ yields a retention half-life of $a_{1/2} = 37$ (Fig. 19). The age–forgetting curve exhibits the familiar two-regime structure — a low-error plateau followed by a sigmoid transition — with no confusion overshoot ($\bar{F}$ saturates at 1 rather than exceeding it). The absence of overshoot is explained by the nature of the daily variation: since only the weights change (not the component means), the replayed means for old days converge toward a time-averaged centroid rather than being actively displaced past it.

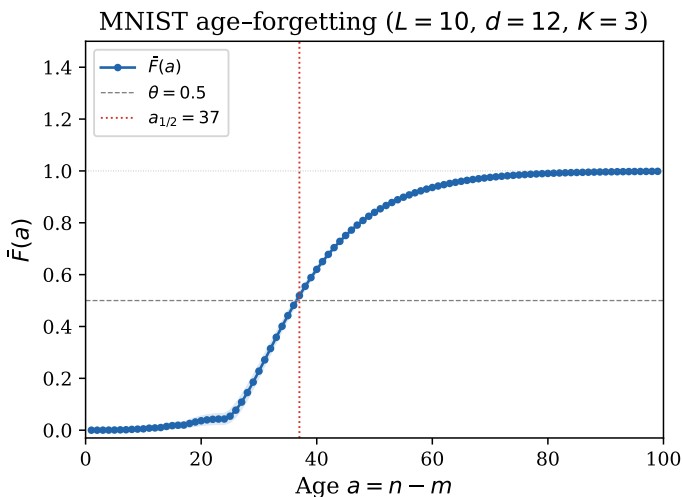

Figure 19: MNIST age–forgetting curve ($L = 10$, $d = 12$, $K = 3$). The half-life $a_{1/2} = 37$ and the curve shape is a clean sigmoid without the confusion overshoot seen in the synthetic mean-drift experiments.

Fig. 20 compares the MNIST and synthetic $K = 3$ forgetting curves at the same $L = 10$ and $P = 30$. The MNIST half-life ($a_{1/2} = 37$) exceeds the synthetic one ($a_{1/2} = 21$). Two effects contribute: (i) the higher latent dimension $d = 12$ inflates the amnesia baseline, diluting the normalized metric; and (ii) the MNIST curriculum perturbs only the weights (a $K$-dimensional vector), while the synthetic curriculum moves all $K$ component means through $\mathbb{R}^2$, producing larger per-step re-binning error.

## 8.3 Decomposed forgetting: covariance-dominated regime

The forgetting decomposition (Fig. 21) reveals a qualitative difference from the synthetic experiments. In the MNIST construction, $F_{\mathrm{cov}}$ dominates the raw forgetting, accounting for the overwhelming majority of the total, while $F_{\mathrm{mean}}$ is comparatively small and $F_{\mathrm{weight}}$ contributes visibly but remains secondary. This is the opposite of the synthetic case (where $F_{\mathrm{mean}} \approx 85\%$) and is explained by the design of the curriculum: since component means are fixed, the mean channel accumulates minimal re-binning drift; instead, the $d \times d$ covariance matrices ($12 \times 12 = 144$ entries per component) accumulate Frobenius-norm error as the piecewise-linear interpolation progressively distorts the class-specific covariance structure. The weight error is non-negligible here because the rotating weights are the primary information channel, unlike the synthetic equal-weight setting.

The raw forgetting also exhibits periodic oscillations at large age (visible in Fig. 21), with period $P = 30$ matching the curriculum. The mechanism is straightforward: with $K = 3$ classes cycling with period $P = 30$, days separated by exactly 30 (or 60, 90, . . . ) share the same dominant digit class. When such a day is recalled, the replayed weight vector happens to be closer to the original (since both have the same class dominant), producing a dip in raw forgetting. Days at half-period offsets ($a = 15, 45, . . .$) have maximally mismatched

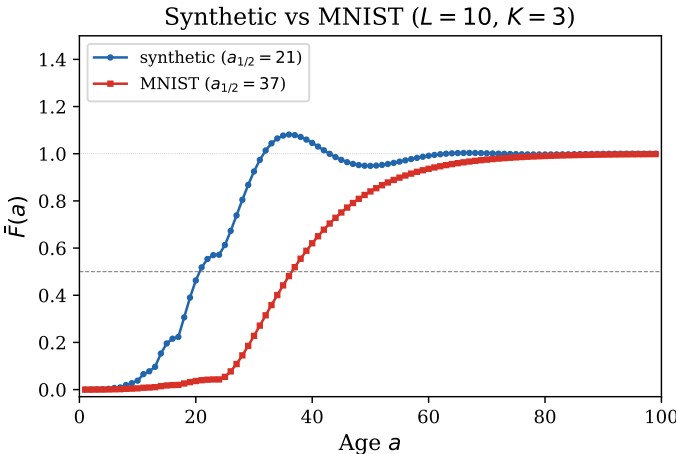

Figure 20: Comparison of synthetic $K = 3$ (blue, $d = 2$, circle drift, $P = 30$) and MNIST (red, $d = 12$, rotating-dominance weights, $P = 30$) age–forgetting curves at $L = 10$. The MNIST curve is shifted rightward due to the higher dimension and milder daily perturbation.

weight vectors and produce forgetting peaks. This resonance effect is purely a consequence of the periodic curriculum and has no analogue in the synthetic mean-drift experiments, where the daily dynamics are continuous rather than weight-modulated.

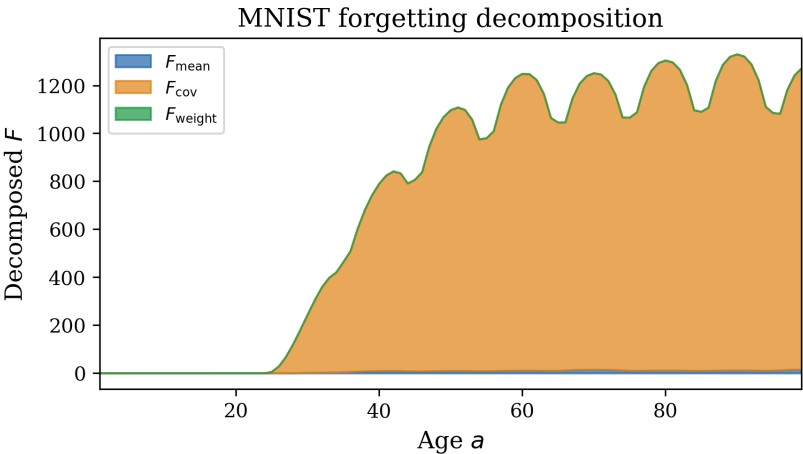

Figure 21: Decomposed raw forgetting for MNIST. Unlike the synthetic experiments where $F_{\mathrm{mean}}$ dominates, the MNIST construction is covariance-dominated because component means are fixed and only the weights rotate. The periodic oscillations at large age have period $P = 30$, matching the curriculum: dips occur at ages that are multiples of $P$ (where the recalled and current days share the same dominant digit class), while peaks occur at half-period offsets.

## 8.4 Visual forgetting and the protocol movie

The key diagnostic of the MNIST experiment is visual: decoded images reveal how forgetting manifests in pixel space.

Fig. 22 shows the per-component replayed means (decoded to $28 \times 28$ pixels) for eight selected past days. For recent days (d90, d99), the three components decode to clearly distinct, recognisable digits (0, 3, 8). As age increases, the components blur and converge: by day 25 and earlier, all three components decode to a similar ambiguous shape resembling the PCA grand mean. This is the visual manifestation of confusion

— not semantic collapse (which would mean instant failure), but progressive loss of class identity through cumulative re-binning.

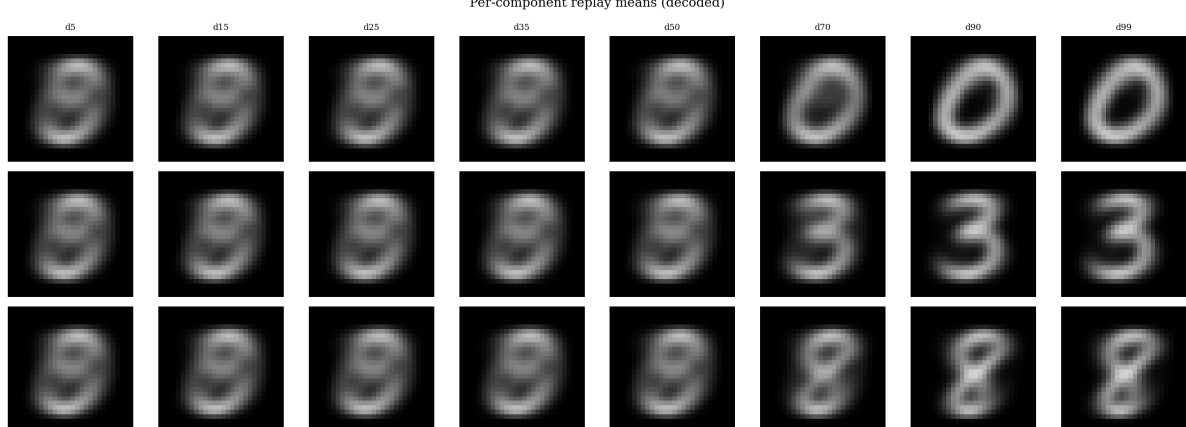

Figure 22: Per-component replayed means decoded to pixel space, for selected past days. Each row corresponds to one digit class (0, 3, 8 from top to bottom). Recent days show distinct digit identities; older days progressively converge toward a common blurred average.

The protocol grid, evaluated at uniformly spaced times $t \in [0, 1]$ and decoded frame-by-frame, produces a *marginal protocol movie* of the compressed history. Fig. 23 shows the per-component strip: each row tracks one digit class's mean through the full protocol. All three digit identities are maintained across the displayed $0 \to 1$ interval — digit 0 remains recognizably 0, digit 3 remains 3, digit 8 remains 8 — although the oldest portion ($t \approx 0$) is visibly blurrier. This figure demonstrates smoothness and semantic persistence of the stored marginal protocol; it should not be read as evidence that simulated SDE trajectories have been validated.

Figure 23: Component movie: per-class decoded means from $t = 0$ (oldest memories, left) to $t = 1$ (current day, right). Each row is one digit class. Digit identities are preserved across the full protocol, with only gradual loss of sharpness at the oldest end.

The protocol weight evolution (Fig. 24) shows the compressed temporal history of the rotating curriculum. At $t = 1$ (current day), digit 8 dominates ($\pi_8 \approx 0.9$). Moving leftward: digit 0 peaked around $t \approx 0.4$, digit 3 around $t \approx 0.2$, and the oldest memories ($t \approx 0$) have roughly equal weights. This weight trajectory is a lossy but recognisable compression of the full 100-day curriculum.

## 8.5 Latent replay classification proxy

The previous MNIST diagnostics evaluate the compressed density protocol itself. We also test a simple task-level consequence of replay: whether the recalled density preserves class-posterior information in the PCA latent space. For each recalled day $m$ at final day $N = 100$, we draw held-out latent samples from the original daily GM and evaluate the class posterior induced by the replayed GM. The classifier is not trained; it is the Bayes classifier obtained from the replayed mixture components associated with digits 0, 3, and 8. We report held-out accuracy and class-posterior negative log likelihood (NLL). The compressed methods all use the same memory budget: CAS uses $L = 10$, hence $L + 1 = 11$ stored protocol states, and each streaming baseline stores $B = 11$ daily GM states. The original daily GM is shown only as an oracle reference, not as a deployable fixed-memory method.

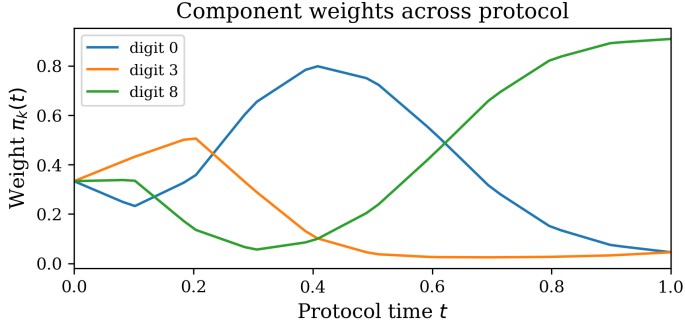

Figure 24: Component weights $\pi_k(t)$ across the protocol grid. The weight trajectory encodes a compressed history of the 100-day rotating-dominance curriculum: recent dominance of digit 8 (green at $t = 1$), earlier dominance of digit 0 (blue peak at $t \approx 0.4$), and convergence toward uniform weights at the oldest end.

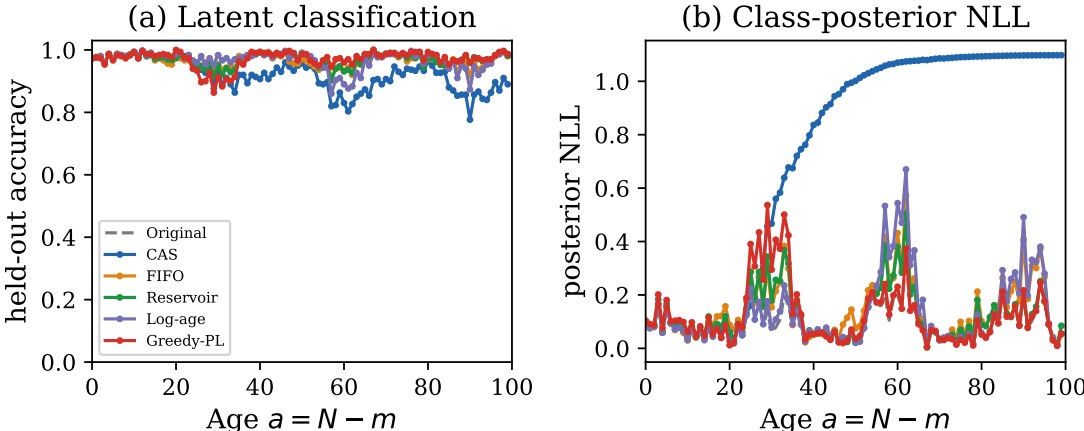

Figure 25: MNIST latent replay classification proxy at final day $N = 100$ under a common $B = 11$ GM-state budget. Panel (a) shows held-out Bayes-classification accuracy under the replayed GM; panel (b) shows the class-posterior NLL. CAS preserves usable class information but is less accurate and less calibrated for old memories than the knot-based streaming baselines.

Fig. 25 and Fig. 26 show that CAS preserves usable but degraded class information. Its all-age accuracy is 0.920 and drops to 0.898 for memories older than 30 days; the corresponding posterior NLL values are 0.732 and 0.998. By contrast, the memory-matched buffers remain closer to the original-GM oracle on this particular task. The strongest method is the greedy piecewise-linear buffer, with all-age accuracy 0.971, old-memory accuracy 0.973, all-age posterior NLL 0.126, and old-memory posterior NLL 0.120. FIFO, reservoir, and log-age buffers are also substantially better than CAS in posterior NLL. Density NLL shows the same ordering: CAS gives 24.07 overall and 26.08 for old memories, whereas reservoir, log-age, and greedy-PL are near 19–19.5.

This task is intentionally favorable to knot-based baselines: the class posterior depends strongly on the rotating mixture weights, and direct storage of selected daily GMs preserves these weights more faithfully than repeated uniform re-binning. The result is therefore not a failure of the density-protocol construction, but a useful diagnostic of what uniform CAS is and is not optimized for. CAS gives a systematic, analytically transparent temporal compression mechanism; it is not an optimal task-specific replay compressor. Task-level losses such as posterior NLL point to the natural next design step: replace uniform re-binning by an adaptive or multi-resolution CAS grid whose knots are chosen by an explicit density or downstream distortion criterion.

MNIST latent replay task, B=11 GM states

| method | states | acc all | acc age≥30 | NLL all | NLL age≥30 |
|--------|--------|---------|------------|---------|------------|
| Original | 0 | 0.979 | 0.979 | 0.090 | 0.092 |
| CAS | 11 | 0.920 | 0.898 | 0.732 | 0.998 |
| FIFO | 11 | 0.966 | 0.964 | 0.147 | 0.160 |
| Reservoir | 11 | 0.970 | 0.969 | 0.127 | 0.131 |
| Log-age | 11 | 0.967 | 0.962 | 0.136 | 0.158 |
| Greedy-PL | 11 | 0.971 | 0.973 | 0.126 | 0.120 |

Figure 26: Aggregate MNIST latent replay task metrics. The original daily GM is an oracle reference. CAS and all streaming baselines use the same $B = 11$ stored GM states. "age$\geq 30$" reports performance on older memories only.

## 8.6 Dimension sweep

Sweeping the PCA dimension $d \in \{4, 8, 12, 20, 30\}$ yields half-lives $a_{1/2} \in \{36, 37, 37, 37, 37\}$ (Fig. 27) — essentially flat across all dimensions tested. No semantic collapse occurs at any $d$. This confirms the dimension-independence observed in the synthetic scaling experiments (Section 7) and demonstrates that the CAS framework handles real image-derived latent spaces as robustly as synthetic ones.

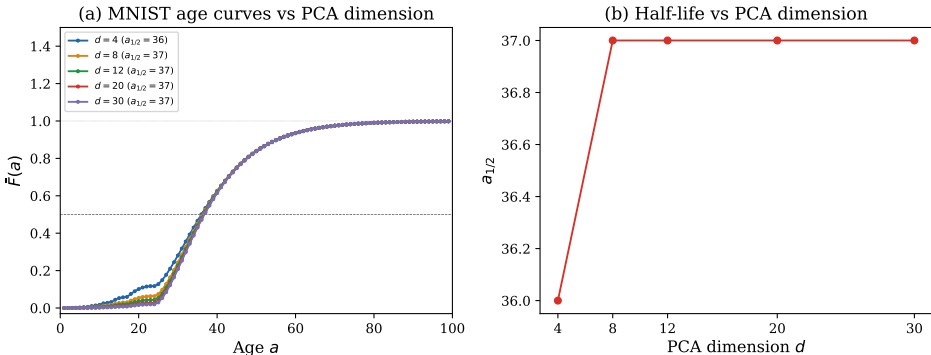

Figure 27: MNIST PCA dimension sweep. (a) Age–forgetting curves for $d \in \{4, 8, 12, 20, 30\}$ nearly coincide. (b) Half-life is flat at $a_{1/2} \approx 37$ across all dimensions.

## 8.7 Takeaway

The MNIST experiment provides an image-derived latent-space illustration of the same density-compression mechanism, together with a simple task-level probe. First, the two-regime forgetting curve persists in this constructed PCA/Gaussian-mixture setting. Second, the dominant error channel differs from the synthetic mean-drift examples: because the component means are fixed and the weights rotate, covariance and weight-related effects become more visible under the present decomposition. Third, decoding the protocol grid frame by frame gives a useful visual diagnostic of marginal memory degradation. Fourth, the PCA dimension sweep does not show semantic collapse over the tested range $d = 4$ to $d = 30$. Fifth, the latent replay classification task shows that uniform CAS is not task-optimal: it preserves nontrivial class information but is outperformed by memory-matched buffers that store or adaptively retain daily GM knots. These conclusions remain limited to the chosen latent representation and diagnostics, and they motivate distortion-aware CAS variants rather than stronger claims about optimal replay.

## 9 Discussion

We now discuss two cross-cutting themes that emerge from the current experiments: the empirical linear retention law observed under the tested diagnostics, and the distinction between the deterministic density protocol and its optional stochastic realization.

## 9.1 Empirical retention law and rate–distortion interpretation

The experiments suggest an approximate law

$$a_{1/2} \approx c\,L, \tag{17}$$

for smooth density streams under the present diagnostics. The important point is the linear dependence on the temporal budget. The coefficient $c$, however, should be treated as source- and metric-dependent. In the single-Gaussian circular experiment, the default setting gives a value near 2–3, while changes in drift speed and geometry shift the observed half-life. Thus the coefficient should not be viewed as a universal capacity constant, but as an empirical compression-efficiency coefficient for a specified source class, representation, and distortion measure.

**Why linear scaling is natural.** For large $L$, the readout time for a memory of age $a$ is approximately

$$t(a) = \left( \frac{L}{L+1} \right)^a \approx e^{-a/L}. \tag{18}$$

If a given source and metric have an effective resolution threshold $t_*$ at which the accumulated rebinning distortion crosses the criterion defining $a_{1/2}$, then

$$a_{1/2} \approx L \log(1/t_*). \tag{19}$$

This argument explains why a linear dependence on $L$ is expected, while also making clear why the proportionality constant can vary with drift speed, geometry, normalization, and metric.

**Relation to rate–distortion.** The CAS protocol is a fixed-rate representation of a distribution-valued time series: $O(LKd^2)$ stored numbers represent a longer stream of daily densities. Re-binning induces distortion. This naturally suggests a rate–distortion viewpoint, where the relevant question is how much temporal history can be retained below a chosen distortion level for a given memory budget. Notice that we use "capacity" only as an analogy and do not claim a Shannon-channel theorem or a metric-independent constant. A formal bound would require specifying the source distribution over density paths, the admissible encoders, and the distortion metric.

This interpretation points to concrete algorithmic extensions:

- *Non-uniform grids*, for example logarithmic grids, to allocate more resolution to recent memories.
- *Variational rebinning*, where node locations or node states are chosen to minimize an explicit distributional distortion between the augmented and compressed protocols.
- *Alternative interpolants and metrics*, such as Wasserstein interpolation and sliced-Wasserstein or held-out likelihood diagnostics, to test whether the observed half-life persists beyond global moment mismatch.

Determining optimal constants for specified source classes and metrics is left as a rate–distortion-style problem for future work.

## 9.2 Deterministic protocol and stochastic replay

The CAS memory stores a density path $p_t(x)$ for $t \in [0, 1]$. This path can be queried directly at any readout time, and all deterministic forgetting diagnostics in this paper are computed from those queried marginals. No SDE trajectory is needed for the memory update or for marginal replay.

Separately, Appendix A reconstructs a drift $s_t(x)$ such that

$$dX_t = s_t(X_t)\,dt + dW_t \tag{20}$$

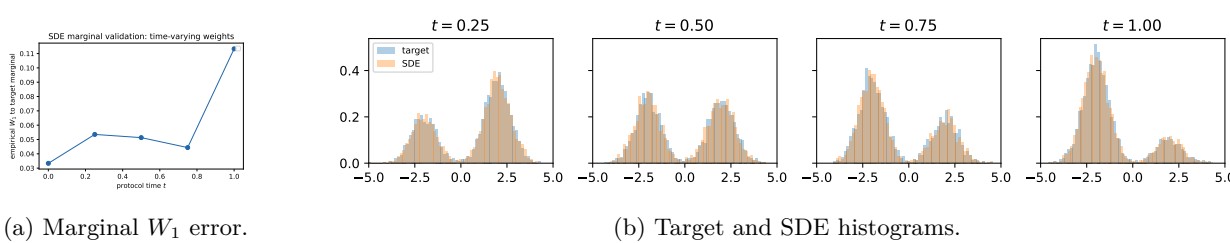

(a) Marginal $W_1$ error.           (b) Target and SDE histograms.

Figure 28: Numerical validation of the time-varying-weight current in a one-dimensional Gaussian-mixture path. This experiment validates the optional SDE realization; it does not alter the deterministic CAS forgetting curves.

has the prescribed one-time marginals $p_t$. This construction provides a possible route to temporally coherent sample paths: samples at different readout times can be coupled along the same stochastic trajectory rather than drawn independently from each marginal.

Therefore we distinguish two notions of replay:

- *Marginal/protocol replay*: evaluate $p_t$ at one or more readout times. This is what CAS directly stores and what the current forgetting curves measure.
- *Trajectory replay*: simulate (20) using the reconstructed drift. This is more expensive, and requires separate numerical validation that simulated marginals match the prescribed $p_t$.

The MNIST frame strips in Section 8 are marginal protocol visualizations. They demonstrate that the stored protocol can be decoded as a smooth visual narrative, but they are not by themselves evidence of validated SDE trajectory replay.

As a numerical validation of the stochastic realization, we consider a one-dimensional two-component Gaussian-mixture path whose component weights vary in time while component means and covariances remain fixed. In one dimension the weight-current can be written equivalently as

$$J_t^{\mathrm{wt}}(x) = -\sum_k \dot{\pi}_k(t)\, \Phi_k(x), \tag{21}$$

where $\Phi_k$ is the Gaussian CDF of component $k$, so that $\partial_x J_t^{\mathrm{wt}} = -\sum_k \dot{\pi}_k(t) g_k(x)$ as required by the continuity equation. Euler–Maruyama simulations using this current match the prescribed marginals: the empirical $W_1$ errors at $t = 0, 0.25, 0.5, 0.75, 1$ are approximately 0.03, 0.05, 0.05, 0.04, and 0.11 (Fig. 28). This validation checks the SDE realization for a nontrivial time-varying-weight density path; it is consistent with the deterministic CAS forgetting curves.

### 9.3 Relation to prior work

Catastrophic interference McCloskey and Cohen (1989) – or catastrophic forgetting French (1999) – in sequentially trained networks has motivated four main CL paradigms: regularization (EWC Kirkpatrick et al. (2017), SI Zenke et al. (2017)), replay (deep generative replay Shin et al. (2017), brain-inspired replay van de Ven et al. (2020)), architecture expansion (progressive nets Rusu et al. (2016)), and compression (Progress & Compress Schwarz et al. (2018)) — all address forgetting-by-interference in shared-parameter models. Our framework is fundamentally different: forgetting arises from temporal coarse-graining rather than parameter overwriting, and the forgetting mechanism is localised in a single identifiable step (re-binning) rather than distributed across gradient updates. Progress & Compress Schwarz et al. (2018) is closest in spirit (it also separates a "knowledge base" from an "active column"), but relies on neural distillation rather than analytical density operations. Variational Continual Learning Nguyen et al. (2018) maintains a sequential posterior, formally similar to our compress–add step; however, it requires gradient-based updates and does not provide a closed-form forgetting analysis. For surveys of the CL landscape, see Parisi et al. (2019); De Lange et al. (2022); Wang et al. (2024).

Within the replay paradigm, recent work replaces the VAE/GAN generator of Shin et al. (2017) with a denoising diffusion model, achieving higher-fidelity replay samples for class-incremental classification Gao and Liu (2023); Jodelet et al. (2023); Meng et al. (2024), object detection Kim et al. (2024), federated learning Liang et al. (2024), industrial streaming data He et al. (2024), and anomaly detection Hu et al. (2025). CAS is not directly comparable to these methods as an end-to-end continual-learning system: it stores a compressed density-valued history and does not train a downstream classifier or policy. The appropriate comparison class for the current paper is therefore memory-matched streaming density summaries, multi-resolution buffers, reservoir-type memories, and online piecewise-linear compression methods; downstream generative replay comparisons require additional experiments.

Our bridge diffusion is constructed by prescribing a density path and recovering the SDE drift from the Fokker–Planck equation (Appendix A). This approach is related to Schrödinger bridges Léonard (2013); Chen et al. (2021); De Bortoli et al. (2021), flow matching Lipman et al. (2023), stochastic interpolants Albergo and Vanden-Eijnden (2023) and Path-Integral Diffusion Behjoo and Chertkov (2025); Chertkov and Behjoo (2025); Chertkov (2025; 2026), but differs in that the density path is specified directly as a piecewise-linear interpolant, rather than optimized or learned.

In the neuroscience literature, Bazhenov and collaborators González et al. (2020); Golden et al. (2022); Tadros et al. (2022); Golden et al. (2025); Vins et al. (2025) show that sleep-like off-line replay can mitigate catastrophic forgetting by interleaving older and newer traces. CAS shares the broad idea of compressed replay, but the present paper studies a density-level compression mechanism rather than synaptic consolidation. Any stronger analogy to biological or RL replay should be made only after trajectory-level SDE replay has been validated experimentally.

## 10 Conclusions and Path Forward

We introduced Compress–Add–Smooth (CAS), a deterministic fixed-budget method for compressing a stream of probability distributions into a piecewise-linear density protocol on $[0, 1]$. In the Gaussian-mixture instantiation, the representation is controlled by a state budget $K$ and a temporal budget $L$, and one update costs $O(LKd^2)$ arithmetic operations. CAS is an analytically transparent model of temporal density compression and forgetting, not yet a complete resource-constrained continual-learning agent.

The current experiments support the following conclusions:

1. *Localized forgetting mechanism.* Compression and addition are lossless or non-destructive at the protocol level; forgetting is localized to the rebinning step that maps an augmented $(L+1)$-segment protocol back to the fixed $L$-segment grid.
2. *Two-regime forgetting curves.* Under the current moment-based diagnostics, smooth Gaussian-mixture streams show a recent-memory plateau followed by a sigmoid-like transition.
3. *Approximately linear retention in $L$.* The half-life grows approximately as $a_{1/2} \approx cL$ in the tested settings. The coefficient $c$ is source-, representation-, and metric-dependent; it should not be interpreted as a universal or Shannon-capacity constant.
4. *Controlled evidence for weak $K$ dependence.* In the labeled Gaussian-mixture experiments, sweeping $K \in \{1, 2, 3, 5, 8\}$ at fixed $L$ produces nearly unchanged moment-based half-lives. This is an empirical observation for the tested streams and metrics, not a general theorem for arbitrary multimodal densities.
5. *Protocol visualization.* The MNIST PCA example shows that the stored marginal protocol can be decoded as a smooth visual strip, but this is distinct from validating actual SDE trajectory replay.
6. *Task-level proxy.* In the MNIST latent replay classification task, CAS preserves usable class-posterior information but is outperformed by memory-matched FIFO, reservoir, log-age, and greedy piecewise-linear buffers. This shows that the uniform CAS grid is a systematic temporal-compression mechanism rather than a task-optimized replay compressor.

Several extensions remain open for future exploration. First, distribution-sensitive diagnostics should be expanded from final-day stress tests to full age-averaged evaluations over multiple random streams. Second, the memory-matched baseline comparison should be broadened beyond the present smooth circular streams

and MNIST latent task; in particular, adaptive and multi-resolution variants should be compared under the same distortion criterion used to train or place their knots. Third, the optional SDE realization should be tested with trajectory-level coherence metrics, not only marginal validation. Fourth, larger downstream proxy tasks or agent tasks are needed before making practical claims about resource-constrained learning systems.

## A  Density interpolants and optional SDE realization

**Scope of this appendix.**  The CAS recursion specifies and updates the density protocol $p_t$ directly. The formulas below are not used by the daily CAS update and are not needed to evaluate replay marginals at readout times. They are needed only if one wants to simulate an Itô process whose one-time marginals follow the prescribed density path. Consequently, the deterministic CAS forgetting curves in this paper are computed from stored protocol marginals, while trajectory-level replay requires separately simulating and validating the SDE realization.

Assume that a smooth density path $p_t(x)$, $t \in [0, 1]$, $x \in \mathbb{R}^d$, is known. We seek a unit-diffusion Itô process

$$dX_t = s_t(X_t)\,dt + dW_t, \tag{22}$$

whose density is exactly $p_t$. The drift must satisfy the Fokker–Planck equation

$$\partial_t p_t(x) + \nabla \cdot J_t(x) = 0, \qquad J_t(x) = s_t(x)p_t(x) - \frac{1}{2}\nabla p_t(x). \tag{23}$$

Once a probability current $J_t$ satisfying the continuity equation is chosen, the drift is

$$s_t(x) = \frac{J_t(x)}{p_t(x)} + \frac{1}{2}\nabla \log p_t(x). \tag{24}$$

### A.1  Gaussian-mixture density paths

Let

$$p_t(x) = \sum_{k=1}^{K} \pi_k(t)g_k(x, t), \qquad g_k(x, t) = \mathcal{N}\big(x; m_k(t), \Sigma_k(t)\big), \tag{25}$$

where $\pi_k(t) > 0$, $\sum_k \pi_k(t) = 1$, and $\Sigma_k(t) \succ 0$.

**Constant weights.**  If $\pi_k(t) \equiv \pi_k$, the component shape current is

$$J_t^{\text{shape}}(x) = \sum_{k=1}^{K} \pi_k g_k(x, t) \left[\dot{m}_k(t) + \frac{1}{2}\dot{\Sigma}_k(t)\Sigma_k(t)^{-1}\big(x - m_k(t)\big)\right]. \tag{26}$$

This current accounts for time-varying component means and covariances.

**Time-varying weights: Poisson construction.**  When the weights vary, write

$$J_t(x) = J_t^{\text{shape}}(x) + J_t^{\text{wt}}(x). \tag{27}$$

The additional source generated by changing weights is

$$r_t(x) = \sum_{k=1}^{K} \dot{\pi}_k(t)g_k(x, t), \qquad \int_{\mathbb{R}^d} r_t(x)\,dx = \sum_{k=1}^{K} \dot{\pi}_k(t) = 0. \tag{28}$$

The correction current must satisfy

$$\nabla \cdot J_t^{\text{wt}}(x) = -r_t(x). \tag{29}$$

Choose $J_t^{\mathrm{wt}} = -\nabla\psi_t$. Then $\psi_t$ must solve

$$\Delta\psi_t(x) = r_t(x). \tag{30}$$

The correct decaying solution is obtained by applying the heat semigroup to the *zero-mass sum $r_t$*, not to each component separately. Define

$$r_{t,s}(x) = \sum_{k=1}^{K} \dot{\pi}_k(t)\,\mathcal{N}\big(x; m_k(t), \Sigma_k(t) + 2sI\big), \qquad s \geq 0. \tag{31}$$

Since $\partial_s r_{t,s} = \Delta r_{t,s}$, $r_{t,0} = r_t$, and $r_{t,s} \to 0$ as $s \to \infty$ because the total mass is zero,

$$\psi_t(x) = -\int_0^\infty r_{t,s}(x)\,ds \tag{32}$$

solves (30). Equivalently,

$$\psi_t(x) = -\frac{1}{(2\pi)^{d/2}} \sum_{k=1}^{K} \dot{\pi}_k(t) \int_0^\infty \frac{\exp\big[-\frac{1}{2}(x - m_k(t))^\top (\Sigma_k(t) + 2sI)^{-1}(x - m_k(t))\big]}{\sqrt{\det(\Sigma_k(t) + 2sI)}}\,ds. \tag{33}$$

The full current and drift are therefore

$$J_t(x) = J_t^{\mathrm{shape}}(x) - \nabla\psi_t(x), \tag{34}$$

$$s_t(x) = \frac{J_t^{\mathrm{shape}}(x) - \nabla\psi_t(x)}{p_t(x)} + \frac{1}{2}\nabla\log p_t(x). \tag{35}$$

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
