# OpenReview forum: "Compress--Add--Smooth: Fixed-Budget Temporal Compression of Density-Valued Streams"
_TMLR — Under review for TMLR_

### Review · Reviewer_RRYs · 2026-05-29

**Summary Of Contributions:**

The paper introduces compress-add-smooth (CAS) update scheme intended for continual learning, especially on resource-constrained edge devices. The strengths includes the simple deterministic update of CAS: interpolation of Gaussian mixture to store past events within $L+1$ grids of memory budgets. However, very little is mathematically or empirically guaranteed about continual-learning performance. While the paper proven e.g., the geometric readout times and and computational complexity under assumptions, but it does not prove a continual learning performance guarantee. Moreover, the paper uses smooth synthetic Gaussian mixture stream and MNIST PCA latent stream, but no downstream tasks are demonstrated.

**Additional Comments:**

Note that the above comments represent my initial impression of the paper, and I am open to discussion. I welcome any corrections to my potential  misunderstandings.

**Audience:**

Yes

**Audience Explanation:**

CAS is well-defined fixed-grid update on labeled Gaussian-mixture parameters. The handled problem of continual learning on resource-restrained device is important, and thus I would say yes to the question "would at least some individuals be interested in the paper?" However, because of the concerns listed above, I would request changes necessary for acceptance, as follows.

**Broader Impact Concerns:**

N.A.

**Claims And Evidence:**

No

**Claims Explanation:**

The paper was an interesting read. Continual learning on resource-constrained devices is an important problem and the CAS framework sounds attractive. However, several concerns prevent me from recommending clear acceptance:

 ## Identifiability of Gaussian mixtures
CAS linearly interpolate component weights, means, and covariances to obtain a density at time $t$. However, a hidden assumption here is the Gaussian mixtures are identifiable, or labeled: permuting components gives the same density. The CAS result can change if the same daily distribution is represented with a different component ordering. The paper may need to define CAS under a space of labeled mixtures to weaken the claim, or introduce a label-alignment algorithm before interpolation.

## Scale-dependence of the forgetting metric (eq. 10)
The two terms in eq. (10) has different units, thus rescaling $x \to \lambda x$ makes mean term scales as $\lambda^2$ while the covariance term scales as $\lambda^4$. The paper does not adequately stress-test robustness to scaling/unit changes.

## Potentially limited novelty
The paper’s exact CAS update for Gaussian-mixture density paths appears distinctive within the CL literature, but the underlying strategy of fixed-budget compression followed by replay/resampling is not novel (e.g., Wang et al., ICLR 2022, “Memory Replay with Data Compression for Continual Learning”). The paper should therefore avoid claiming novelty at the level of “compression-resampling continual learning,” and should compare against the most relevant compression/replay baselines under a true fixed memory budget.

## Experimental evidence is too narrow
Related to the above concern, the paper does not effectively position the proposed method on top of state-of-the-arts. Specifically, the paper uses smooth synthetic Gaussian/Gaussian-mixture streams and a constructed MNIST PCA latent stream, but there is no downstream task, no learned representation, no sample-based fitting cost, and no comparison to serious online compression/replay baselines.


## Potential mistake in Appendix A
Eq. (26) may solve the wrong Poisson equation. Eq. (25) plus $J_t^{\mathrm{wt}}=-\nabla\psi_t$ requires $\Delta\psi_t=f_t:=\sum_k\dot\pi_k g_k$. But the proposed formula $\psi_t=\frac12\int_0^\infty e^{s\Delta}f_t,ds$ gives $\Delta\psi_t=-f_t/2$, since $\partial_s e^{s\Delta}f_t=\Delta e^{s\Delta}f_t$. Consequently the current has divergence $+f_t/2$, not the required $-f_t$. The corrected formula should be $\psi_t=-\int_0^\infty e^{s\Delta}f_t,ds$, with the full zero-mass sum inside the integral.

**Requested Changes:**

## Add strong baselines under continual learning downstream tasks
e.g., FIFO, reservoir sampling, exponential moving average, online spline/piecewise-linear compression, adaptive knot placement, Kalman/state-space summaries, and possibly coreset methods.

## At least reframe claims
As is, CAS is an online temporal density-compression method based on labeled Gaussian-mixture parameters, not yet a general continual-learning solution.

## Integrity of Appendix A
Please double check and correct Appendix A if needed, especially the time-varying-weight Poisson construction, and clearly distinguish the CAS compression algorithm from the optional SDE coupling.

---

> ### Author Response · Authors · 2026-06-24
> **Response to reviewer comments**
>
> We thank the reviewer for the constructive assessment. We agree with the central point that CAS should be framed as an online temporal density-compression method for labeled Gaussian-mixture parameters, not as a general continual-learning solution. This is now the organizing viewpoint of the revised paper.
>
> \textbf{Framing.}
> We revised the title, abstract, introduction, discussion, and conclusion. The paper now presents CAS as a fixed-budget temporal compression scheme for density-valued streams. We explicitly state that the contribution is an analytically transparent model of forgetting by temporal coarse-graining, not a complete replacement for neural continual-learning systems. The phrase ``resource-constrained agent'' is now used only as motivation for the abstraction, not as a claim of demonstrated end-to-end agent performance.
>
> \textbf{Labeled mixtures and identifiability.}
> We agree that componentwise interpolation is not permutation invariant. We now define the Gaussian-mixture version of CAS on labeled mixtures. We also state that if daily mixtures are obtained from unlabeled fitting, then a component-alignment step is required before interpolation, for example Hungarian matching or entropic optimal transport between adjacent days. The decomposed metric already uses Hungarian matching; the algorithmic assumption is now stated clearly in the memory representation section.
>
> \textbf{Metric scale and distributional diagnostics.}
> We agree that the original moment metric has scale and information-loss limitations. We retained it as a cheap analytic diagnostic, but added density-level diagnostics: normalized sliced-Wasserstein distortion and held-out negative log likelihood,
> \[
> D_{\rm NLL}(m,n)=
> \mathbb E_{X\sim p^{(m)}}[-\log \hat p^{(m|n)}(X)+\log p^{(m)}(X)].
> \]
> The new results show that the conclusions are metric-dependent. In particular, held-out NLL gives shorter half-lives than the moment metric. We therefore removed claims of metric-independent retention scaling.
>
> \textbf{Appendix A.}
> We corrected the time-varying-weight Poisson construction. The corrected source is
> \[
> r_t(x)=\sum_k\dot\pi_k(t)g_k(x,t),
> \]
> and the potential satisfies
> \[
> \Delta\psi_t=r_t,\qquad
> \psi_t(x)=-\int_0^\infty r_{t,s}(x)\,ds.
> \]
> We also emphasize that deterministic CAS does not require the SDE construction. The SDE realization is optional and only relevant when one wants coupled trajectory replay rather than marginal/protocol replay.
>
> \textbf{Baselines.}
> We added memory-matched baselines: FIFO, reservoir/uniform-history storage, log-age/multiresolution storage, and online greedy piecewise-linear compression. Memory is reported in the same unit for all methods, namely stored GM states. CAS with \(L=10\) stores \(L+1=11\) GM states, and each baseline receives \(B=11\) states. These comparisons make the story more precise: CAS is systematic and analytically transparent, but not optimal; adaptive and smoothness-aware baselines can outperform uniform CAS on smooth streams.
>
> \textbf{Proxy downstream task.}
> We added an MNIST latent replay task. For each replayed density, we evaluate held-out class-posterior accuracy and NLL. CAS preserves useful class information, but the best memory-matched adaptive baselines perform better. This supports the revised paper thesis: CAS is a methodology-building block and controlled baseline for temporal density memory, while optimized replay and richer generative-AI mechanisms are important future work.

---

### Review · Reviewer_ZBaV · 2026-06-16

**Summary Of Contributions:**

The paper introduces Compress–Add–Smooth (CAS), a fixed-segment representation of historical Gaussian-mixture distributions. New distributions are appended after temporally compressing the existing protocol, followed by rebinning to restore the segment budget. The construction is simple, interpretable, and inexpensive for small Gaussian mixtures. The experiments clearly illustrate how repeated rebinning produces age-dependent degradation. However, the paper’s central claims about universal retention scaling, state-complexity independence, information-theoretic capacity, and stochastic replay are not adequately supported and are sometimes contradicted by the presented results.

**Additional Comments:**

The manuscript is generally clear and visually polished, and releasing code is valuable. In its current form, however, the mathematical issue in Appendix A and the metric-dependent experimental conclusions prevent acceptance.

**Audience:**

Yes

**Audience Explanation:**

An analytically transparent model of lossy temporal memory could interest researchers in continual learning, streaming density estimation, and resource-constrained inference. Localizing forgetting to a particular compression operation is useful, and the CAS recursion is easy to analyze and implement. The current manuscript would be substantially more compelling if positioned as an online temporal density-compression method rather than as a general replacement for continual learning, and if related to existing streaming and multiresolution compression literature.

**Broader Impact Concerns:**

None.

**Claims And Evidence:**

No

**Claims Explanation:**

The basic CAS recursion and its O(LKd^2) protocol update are explained clearly. The evidence does not, however, support the broader conclusions.

First, Appendix Equation (26) appears mathematically incorrect. Let g_s denote the Gaussian with covariance Sigma+2sI. Since partial_s g_s = Delta g_s, Delta integral_0^infinity g_s ds = -g_0. The positive one-half integral in Equation (26) therefore gives -g/2, rather than solving the stated equation Delta psi = g. Consequently, the proposed current for time-varying weights does not satisfy Equation (25). This directly affects the MNIST setting and the claimed SDE replay construction.

Second, the claimed universal law a_1/2 approximately 2.4L is not established. At the default L=10, the reported half-life 30 corresponds to c=3.0, while MNIST gives c=3.7. Circle and linear drift give half-lives 30 and 42, respectively; drift speed changes the half-life from 20 to 36; and crowding changes it from 30 to 20. These are material dependencies, contradicting statements that c is geometry-independent or unchanged across settings. The experiments use a 100-day deterministic sequence, with no independent trials or uncertainty estimates. Large-L half-lives are also estimated from relatively few old-age pairs.

Third, the evaluation metric builds in several reported conclusions. The primary metric reduces every mixture to its global mean and covariance, so it cannot detect changes between multimodal distributions having identical first two moments. The apparent independence from K is therefore not convincing. The decomposed metric adds mean, covariance, and weight errors with incompatible scales; covariance error naturally grows with d^2. Its “channel shares” cannot consequently establish which information channel dominates. Dimension independence is also partly an artifact of the normalization: the paper explicitly notes that the amnesia denominator increases with dimension.

Fourth, FIFO is the only baseline. CAS exploits temporal smoothness, whereas FIFO does not, making the claimed 2.4x capacity improvement source-dependent rather than an information-capacity result. Comparisons against memory-matched streaming summaries, exponential or multiresolution buffers, reservoir sampling, and online piecewise-linear compression are needed. No downstream control, prediction, classification, or replay task demonstrates that lower moment mismatch produces useful behavior.

Fifth, the stochastic-process contribution is not experimentally demonstrated. Figures 19–20 decode protocol parameters or independently evaluated marginals, rather than showing a trajectory sampled from the proposed SDE. The linked code repository, https://github.com/mchertkov/CAS-Bridge-Diffusion, likewise constructs marginal samples and parameter strips but does not implement or validate the stated Poisson correction and SDE trajectory. Thus the “temporally coherent movie replay” claim remains hypothetical.

Finally, the implementation maintains one readout time per historical day and updates all entries daily. This requires O(n) memory and time, conflicting with the fixed-memory and O(LKd^2)-per-day claims. Equation (9) could eliminate this dictionary, but the algorithm and implementation should do so explicitly. Componentwise Gaussian-mixture interpolation also assumes persistent component labels; the synthetic split-and-merge experiment retains fixed identities and does not test genuine component permutation, birth, or death.

**Requested Changes:**

1. (Critical) Correct the time-varying-weight Poisson derivation and verify numerically that simulated SDE marginals agree with the prescribed density path.

2. (Critical) Evaluate replay using distribution-sensitive, permutation-invariant metrics such as Wasserstein distance, held-out log likelihood, or task-specific loss. Normalize mean, covariance, and weight errors before interpreting their relative contributions.

3. (Critical) Reassess the scaling claims using longer horizons, multiple random streams and seeds, confidence intervals, and explicit model fits. Report c as source- and metric-dependent unless a theorem establishes otherwise.

4. (Critical) Add memory-matched streaming baselines that exploit the same smoothness assumptions, along with at least one downstream task demonstrating practical value.

5. (Critical) Provide actual SDE trajectory experiments, not only decoded marginal means, and measure temporal coherence against independent marginal sampling.

6. (Critical) Remove the growing readout dictionary from the production algorithm and account for all memory and computational costs. Address component-label ambiguity using a permutation-invariant update mechanism.

7. (Minor) Temper or remove the “Shannon channel capacity,” “fully analytical Ising model,” and universal-law language unless formal definitions and bounds are supplied.

8. (Minor) Add microcontroller timing, memory, and numerical-stability measurements before claiming hardware viability.

9. (Minor) Discuss prior work on streaming summaries, online time-series compression, multiresolution histories, and piecewise-linear approximation.

---

> ### Author Response · Authors · 2026-06-24
> **Response to Reviewer comments**
>
> We thank the reviewer for the detailed and technically precise review. We agree that the original manuscript made claims that were too broad relative to the evidence, and we revised the paper substantially.
>
> \textbf{Appendix A and SDE replay.}
> We thank the reviewer for identifying the error in the time-varying-weight Poisson construction. We corrected Appendix A. The weight source is now
> \[
> r_t(x)=\sum_k \dot\pi_k(t)g_k(x,t), \qquad \int r_t(x)\,dx=0,
> \]
> and the correction current satisfies
> \[
> \nabla\cdot J^{\rm wt}_t(x)=-r_t(x).
> \]
> With \(J^{\rm wt}_t=-\nabla\psi_t\), the corrected Poisson equation is
> \[
> \Delta\psi_t(x)=r_t(x),
> \]
> with solution
> \[
> \psi_t(x)=-\int_0^\infty r_{t,s}(x)\,ds,
> \quad
> r_{t,s}(x)=\sum_k \dot\pi_k(t)\mathcal N(x;m_k(t),\Sigma_k(t)+2sI).
> \]
> We also clarify that this correction affects only SDE trajectory replay. It does not affect the deterministic CAS forgetting curves, which are computed from stored/replayed density marginals and do not simulate the SDE.
>
> We added a numerical SDE validation for a one-dimensional two-component mixture with time-varying weights. Euler--Maruyama simulations using the corrected current match the prescribed marginals, with empirical \(W_1\) errors approximately \(0.03,0.05,0.05,0.04,0.11\) at \(t=0,0.25,0.5,0.75,1\).
>
> \textbf{Scaling and universality.}
> We agree that the original language about a universal law \(a_{1/2}\approx 2.4L\), independence from \(K,d\), and Shannon capacity was too strong. We revised the abstract, introduction, discussion, and conclusion. We now state that the coefficient in
> \[
> a_{1/2}\approx cL
> \]
> is source-, representation-, and metric-dependent. The rate--distortion analogy remains as motivation, but we no longer claim a Shannon-channel theorem or metric-independent capacity constant.
>
> \textbf{Metrics.}
> We agree that the moment metric is insufficient by itself. We added sliced-Wasserstein and held-out NLL diagnostics. These show that moment and sliced-Wasserstein half-lives are similar in the smooth synthetic streams, while held-out NLL is stricter. This result is now discussed explicitly as metric dependence, not universality.
>
> \textbf{Baselines.}
> We added memory-matched baselines with memory reported in identical units: stored GM states. CAS with \(L=10\) stores \(L+1=11\) GM states, and all baselines are given the same budget \(B=11\). We compare against FIFO, reservoir/uniform-history storage, log-age/multiresolution storage, and online greedy piecewise-linear compression. The results show that CAS improves over naive FIFO under the original forgetting metric, but adaptive or smoothness-aware baselines can outperform uniform CAS. We now present this honestly: CAS is not claimed to be optimal.
>
> \textbf{Downstream/proxy task.}
> We added an MNIST latent replay task measuring held-out class-posterior accuracy and NLL under replayed densities. CAS preserves useful class information but is weaker than the best memory-matched adaptive baselines. This supports the revised positioning of CAS as a principled and analyzable temporal-compression baseline, not a fully optimized continual-learning method.
>
> \textbf{Fixed memory and labels.}
> We removed the production \(O(n)\) readout dictionary. A deployed memory computes \(t(a)=(L/(L+1))^a\) on demand. We also made the labeled-mixture assumption explicit and state that unlabeled fitted mixtures require component alignment, e.g. Hungarian matching or entropic optimal transport, before interpolation.

---

### Review · Reviewer_Yjym · 2026-06-19

**Summary Of Contributions:**

**Summary**
The authors propose to realize memory for replay through a stochastic process they call Bridge Diffusion. The framework is equipped with $L+1$ "storage units" (nodes) placed at uniform times $\{0, 1/L, \dots, 1\}$ on a replay interval $[0,1]$, where $L$ is a hyperparameter controlling the temporal resolution (and $K$ controls the number of mixture components per node). Each node holds a marginal distribution — parameterized as a Gaussian mixture — that summarizes the experiences of one past "day". Between nodes, the marginals are obtained by piecewise-linear interpolation of the Gaussian-mixture parameters, and the Bridge Diffusion is the stochastic process whose time-marginals follow this prescribed density path (its drift is reconstructed from the Fokker–Planck equation rather than learned). New experiences are integrated by a three-step recipe (Compress–Add–Smooth): a) compress, i.e. rescale the time axis from $[0,1]$ to $[0, L/(L+1)]$, contracting the existing nodes and freeing space near $t=1$; b) add the new day's distribution as a new node at $t=1$, yielding $L+2$ nodes; and c) smooth, i.e. rebin the $L+2$ nodes back to $L+1$ nodes on the uniform grid. The authors perform extensive evaluations of the memory's forgetting behavior on toy data using Gaussian mixtures and MNIST. The authors state that, although only Gaussian mixtures are used in the experiments, their framework applies to any parameterised density family for which piecewise-linear interpolation is well defined.

**Strengths**
* **(S1 - originality and interest to the community)**. The proposed approach is interesting and original, and should appeal to part of the TMLR audience. Together with **(S2, S3)**, the memory's design is generally convincing. I like the idea of realizing storage through bridge diffusion, and I also like the merged replay "movie" across the different marginal distributions. The plateauing effect for recent experiences also seems like a property potentially well aligned with the real-world needs of agent memories.
* **(S2 - mathematical foundation)**. The mathematical derivation appears sound and provides a solid foundation for the proposed framework; realizing the memory through the described stochastic process makes sense. (A few symbols are used before being defined — see W4 — but the core derivation holds.)
* **(S3 - extensive study of forgetting)**. The extensive study of the forgetting behavior is appealing and provides various insights, e.g. regarding the retention half-life.

**Weaknesses**
* **(W1 - lack of practical experiments and comparison; claims vs. evidence)**. All experiments evaluate the forgetting behavior of the memory in isolation. While this is interesting, it leaves the manuscript's core motivation — memory for resource-constrained ML *agents* — unsupported by evidence. Under TMLR's criteria the cleanest resolution is a disjunction: either demonstrate the memory in a downstream agent setting, or adjust the framing to "an analytically tractable model of forgetting" and scale back the edge-agent claims. Concretely: **(Q1):** How do agents equipped with the designed memory perform compared to other bounded-memory methods in the field? State-space models, for instance, also have bounded memory — how would the diffusion-bridge agents compare against them, both performance-wise and in hardware/resource terms? **(Q2):** Most RL agents update their policy by computing gradients on the instance level. On a practical note, how would you form the per-day marginal experience distribution? The MNIST experiment seems quite far from reality, as it assumes the agent sees only one type of digit per day. And if the experience distribution is realized with a Gaussian mixture, how much information is lost compared to instance-based memory systems? **(Q3):** How does the approach scale with the experience dimension $d$? Each node stores $d \times d$ covariances ($O(LKd^2)$), and Gaussian mixtures in high-dimensional raw spaces are weak — what happens well beyond MNIST-scale dimensions?

* **(W2 - lack of comparison against other generative approaches for replay)**. The authors briefly mention some other generative approaches in the related-work section, but an experimental comparison is missing and would strengthen the evidence. **(Q4):** Could a comparison against representative generative-replay baselines be included?

* **(W3 - writing and manuscript quality)**. While the experimental sections are well written and easy to follow, the first part of the manuscript is a hard read for anyone without a background in stochastic processes and probabilistic modellng. The early usage of terms like "terminal marginal" or "bridge diffusion" without explanation diminishes the accessibility of the paper for a wider audience. The authors might consider restructuring the manuscript and including a background section that introduces the problem setting and proactively defines concepts needed later, e.g. bridge diffusion (as partly done in the Appendix) (**Q5**).

* **(W4 - double-blind review process broken)**. The submission includes an acknowledgments section and a GitHub link, which reveal the authors' identities and break the double-blind review process. These must be removed for an anonymized submission. **Note for AE**: As I do not recognize the name provided in the acknowledgment section, and as I did not search for more information, I still feel unbiased in the sense that I believe I can provide a fair review.

**TLDR**
The manuscript is technically solid and its innovative memory design is of interest to part of the TMLR community. However, its central, practically framed claims about resource-constrained agents are not yet matched by evidence: the experiments measure forgetting in isolation.

**Audience:**

Yes

**Audience Explanation:**

See **(S1)**.

**Broader Impact Concerns:**

No concerns.

**Claims And Evidence:**

No

**Claims Explanation:**

See **(W1)**.

**Requested Changes:**

* Requested changes / things to address: Please see **(Q1)**, **(Q2)**, **(Q4)**.
* Optional: **(Q3)**, **(Q5)**.

---

> ### Author Response · Authors · 2026-06-24
> **Response to Reviewer comments**
>
> We thank the reviewer for the positive assessment of the core idea and for clearly identifying the gap between the original practical framing and the evidence. We followed the reviewer's suggested resolution: rather than trying to claim a full downstream agent system, we reframed the paper as an analytically tractable model of density-level temporal memory and forgetting.
>
> \textbf{Claims versus evidence.}
> The revised manuscript no longer presents CAS as a general continual-learning solution for resource-constrained agents. We now describe the method as a deterministic fixed-budget compression scheme for streams of probability distributions. The continual-learning and edge-agent motivations remain, but the demonstrated contribution is narrower: localization and measurement of forgetting caused by temporal coarse-graining.
>
> \textbf{Problem setting and readability.}
> We rewrote the abstract and introduction to define the problem before introducing bridge diffusion terminology. We now first introduce a stream of daily distributions
> \[
> q^{(1)},q^{(2)},\ldots,q^{(n)}
> \]
> and the goal of maintaining a fixed-size density-valued memory. Only after this do we introduce the CAS protocol on \([0,1]\). We also separate deterministic CAS from the optional stochastic realization. This should make clear that the daily memory update is not an SDE simulation.
>
> \textbf{Marginal replay versus trajectory replay.}
> We now distinguish two notions of replay. Marginal/protocol replay means evaluating the stored density path \(p_t\) at one or more readout times; this is what CAS directly stores and what the forgetting curves measure. Trajectory replay means simulating an SDE whose marginals follow \(p_t\); this is optional and requires separate validation. The MNIST visual strips are now described as marginal protocol visualizations, not as proof of validated SDE trajectory replay.
>
> \textbf{Practical experiments and baselines.}
> We added memory-matched baseline comparisons. CAS with \(L=10\) stores \(L+1=11\) GM states, and FIFO, reservoir/uniform-history, log-age/multiresolution, and online greedy piecewise-linear baselines all receive the same \(B=11\)-state budget. The results show that CAS is better than naive FIFO in the controlled forgetting metric, but that adaptive/smoothness-aware baselines can outperform uniform CAS. This supports the revised framing: CAS is a simple, analyzable baseline, not an optimized compression method.
>
> \textbf{MNIST latent proxy task.}
> We added a proxy task on the MNIST PCA latent representation. We evaluate replayed densities by held-out class-posterior accuracy and NLL. CAS retains usable semantic information, but the adaptive baselines are stronger. This result is valuable because it identifies the next algorithmic direction: replacing uniform CAS rebinning by adaptive or distortion-aware temporal compression, and eventually by richer generative-AI replay mechanisms.
>
> \textbf{Scaling and high-dimensional representations.}
> We clarified that the current experiments do not establish scaling to arbitrary high-dimensional raw observations. The MNIST example is explicitly a PCA/Gaussian-mixture latent-space illustration. We now state that richer learned representations, normalizing flows, score models, or inference-time generative replay mechanisms are natural future work.
>
> \textbf{Double-blind issue.}
> We removed the acknowledgments and anonymized identifying code references from the double-blind manuscript.
>
> Overall, we believe the revised manuscript now makes a more precise and defensible contribution: CAS is the simplest systematic density-protocol memory on which one can study forgetting by compression. It is not claimed to be optimal; rather, it provides the controlled baseline from which adaptive and generative extensions can be developed.